# Network-based machine learning in colorectal and bladder organoid models predicts anti-cancer drug efficacy in patients

JungHo Kong[1], Heetak Lee [1], Donghyo Kim[1], Seong Kyu Han[1], Doyeon Ha[1], Kunyoo Shin [1,2 ✉] & Sanguk Kim [1,2 ✉]

Cancer patient classification using predictive biomarkers for anti-cancer drug responses is essential for improving therapeutic outcomes. However, current machine-learning-based predictions of drug response often fail to identify robust translational biomarkers from pre-clinical models. Here, we present a machine-learning framework to identify robust drug biomarkers by taking advantage of network-based analyses using pharmacogenomic data derived from three-dimensional organoid culture models. The biomarkers identified by our approach accurately predict the drug responses of 114 colorectal cancer patients treated with 5-fluorouracil and 77 bladder cancer patients treated with cisplatin. We further confirm our biomarkers using external transcriptomic datasets of drug-sensitive and -resistant isogenic cancer cell lines. Finally, concordance analysis between the transcriptomic biomarkers and independent somatic mutation-based biomarkers further validate our method. This work presents a method to predict cancer patient drug responses using pharmacogenomic data derived from organoid models by combining the application of gene modules and network-based approaches.

[1] Department of Life Sciences, Pohang University of Science and Technology, Pohang 790-784, Korea. [2] Institute of Convergence Science, Yonsei University, Seoul 120-749, Korea. ✉email: kunyoos@postech.ac.kr; sukim@postech.ac.kr

dentifying molecular biomarkers for classification of cancer patients according to drug sensitivity is crucial for the successful treatment of cancer patients[1–3]. Although biomarkers associated with drug response are often found in patient cohort data, clinical trials are still extremely expensive and time-consuming to conduct[4]. Therefore, the accurate discovery of robust biomarkers from preclinical models, which are more accessible than clinical data, is becoming increasingly important. Accordingly, previous findings from large-scale pharmacogenomic screenings of preclinical models have been extremely useful in the discovery of clinically relevant biomarkers. Moreover, studies have reported that machine-learning (ML) algorithms trained from preclinical model data were predictive of cancer patient drug responses[1], which further supports the use of preclinical models for understanding the therapeutic responses of human cancers[3,5–7].

However, preclinical models for drug biomarker identification or ML model development frequently fail to predict drug sensitivity in human tumors[8,9]. Differences in the complexity of the biological systems are one challenge of these models[10,11]. Also, limited training data can hinder the performance of ML techniques, in contrast to data-rich input features, such as gene expression profiles. Input feature complexity, also known as input heterogeneity, poses key challenges in most biological studies, including drug-response prediction tasks, in which drug screening results are scarce compared to the density of high-throughput sequencing data. Therefore, a method to reduce biological heterogeneity and to select relevant features, while developing an efficient model for ML, is required to make robust predictions.

Network-based methods offer a powerful framework to successfully enable feature selection, which in turn may be leveraged to develop robust ML techniques for drug-response prediction. Previous studies have found that genes that are associated with similar phenotypes are in proximity to each other in protein–protein interaction (PPI) networks[12,13], suggesting that drug-response-associated biomarkers may also cluster within specific PPI networks. In addition, Guney et al. demonstrated that the therapeutic effect of a drug could be inferred from the drug-disease proximity within a PPI network, and Fernández-Torras et al. showed that gene modules in a network could be used to predict drug response[7,14]. Furthermore, Cheng et al. applied a network approach to generate the framework for identifying drug repurposing candidates for 220 million patients[15]. Recently, this work was expanded on using network analysis for drug repurposing for cancer patients[16]. Altogether, these findings demonstrate that network-based approaches can be utilized to reduce biological complexity and to improve the performance of ML methods to predict therapeutic outcomes in cancer patients.

Three-dimensional (3D) organoid culture models are being actively developed to improve the pharmacogenomic similarities between preclinical models and human tumors. A recent study found that the transcriptomes of organoid models closely resembled those of human metastatic breast cancer[17]. In addition, Vlachogiannis et al. observed that drug sensitivity data from organoid models were similar to the drug responses observed in gastrointestinal cancer patients[18]. Similarly, Ooft et al. discovered that organoid models were predictive of colorectal cancer patient responses to irinotecan-based chemotherapy[19]. Thus, because of the high similarity between organoid models and human tumors, methods to culture and screen organoid models in an automated, high-throughput manner are actively being developed[20]. Taken together, organoid models recapitulate human tumors at molecular and phenotypic levels, further supporting their use in drug biomarker discovery. However, a method to systematically identify biomarkers from organoid models to predict drug responses in cancer patients remains an unmet need.

In this study, we integrated pharmacogenomic data derived from 3D organoid culture models and network-based methods to develop an ML framework for the prediction of patient–drug responses. Specifically, prior to ML, we conducted a feature selection procedure in which features (pathways) were selected by their proximity to drug targets in a PPI network. We then used the expression profiles of the pathways proximal to the drug targets from organoid models to train ML models and to obtain drug-response biomarkers. To test the predictive performance of the ML model, we analyzed the drug treatment responses of colorectal and bladder cancer patients to 5-fluorouracil (5FU) and cisplatin, respectively. The biomarkers identified by our method were predictive of overall survival in both the colorectal and bladder cancer patients, whereas conventional ML models using whole-genome or whole-pathway transcriptomics were not strong predictors of overall survival. We further confirmed our biomarkers in independent transcriptomic datasets of drug-sensitive and -resistant isogenic cancer cell lines and found that the differential expression patterns were consistent with our network model predictions. Furthermore, predicted biomarkers from our models were similar to somatic mutations of clinically identified independent biomarkers. To the best of our knowledge, this work presents the first systematic framework to leverage organoid pharmacogenomic data and network-based computational approaches to obtain robust drug biomarkers for successful treatment of human tumors.

## Results

**Identification of biomarkers associated with anti-cancer drug responses using a network-based machine-learning approach.** Our approach integrates pharmacogenomic screening in 3D organoid culture models with network-based analysis to provide molecular signatures associated with drug responses. Existing drug-response prediction methods often rely solely on molecular signatures, such as transcriptomic data, to generate predictive models and/or to identify drug-response biomarkers. In contrast, we applied PPI networks and known drug targets to identify potential biological pathways that are associated with drug responses (Fig. 1a). Distances between biological pathways and known drug targets within a PPI network were computed, and pathways that were more proximal than random expectations were considered as potential drug biomarkers (Fig. 1b.1). Here, we used the STRING PPI network, which is comprised of 13,824 proteins and 323,774 interactions[21]. Of note, the STRING network is an undirected network and therefore does not have "upstream" or "downstream" directionality. To train the ML model, we used recent pharmacogenomic screening data from 3D organoid models for 19 colorectal cancer[22] and nine bladder cancer organoid samples[23]. We used ridge regression as our primary ML model, but we also used linear and support vector regression to test it. Expression profiles of inferred biomarker pathways were used as the input data to train the ML model against drug-response measurements (IC$_{50}$, the half-maximal inhibitory concentration) from organoid models (Fig. 1b.2). Finally, pathways with high predictive performance in the ML model were considered to be drug-response biomarkers (Fig. 1b.3). To validate the identified drug-response biomarkers, we measured and compared the clinical benefits of overall survival in the predicted responders and non-responders, who were classified according to transcriptomic levels of the identified biomarkers (Fig. 1b.4).

**Identification of biomarkers in organoid models.** We first applied our model for the identification of drug-response biomarkers following 5FU treatment in colorectal cancer. Among

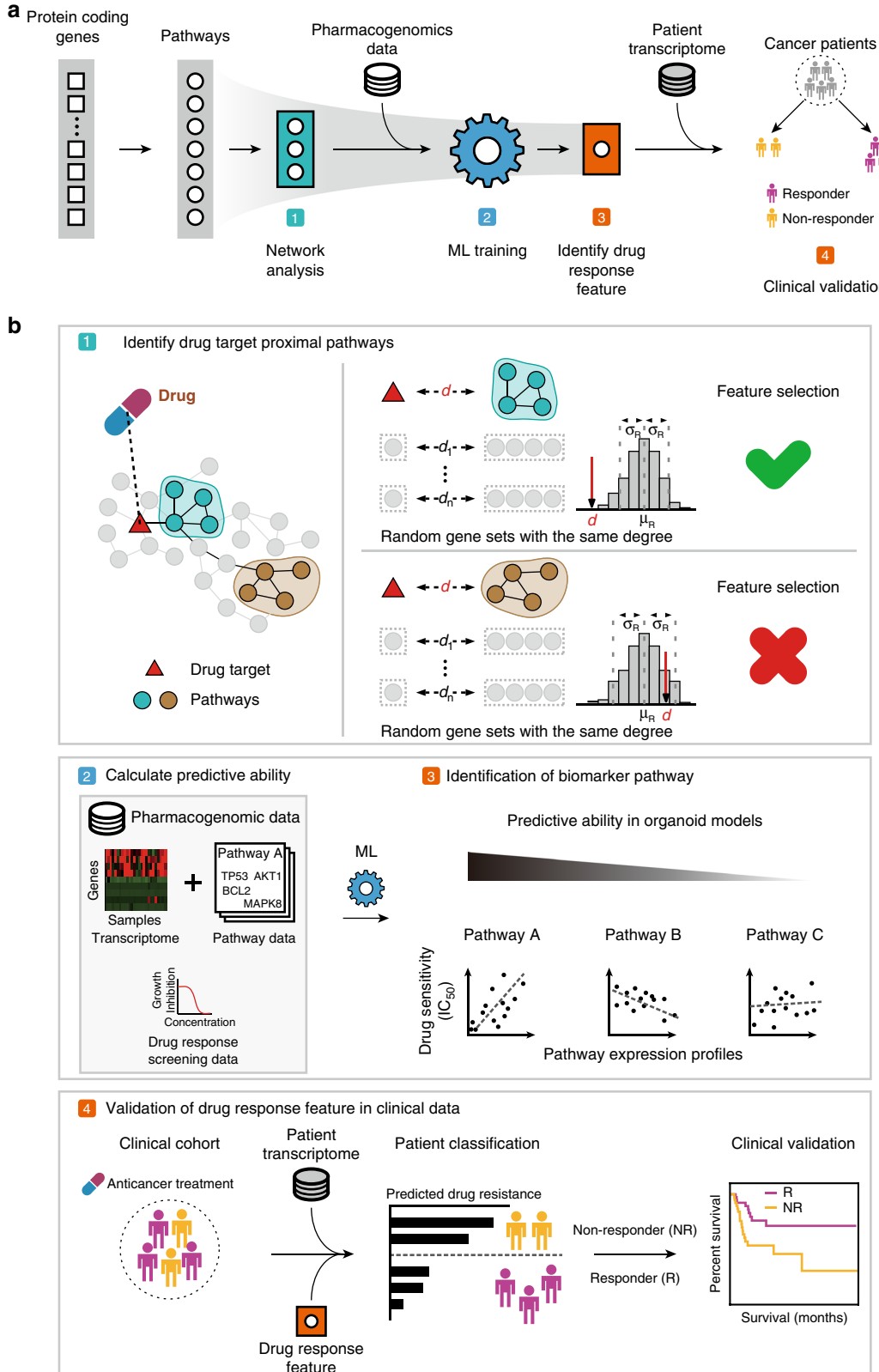

674 reactome pathways, we found 37 pathways that were proximal to the 5FU target in the PPI network (Fig. 2a). To measure the predictive performance of the proximal pathways against the 5FU drug response in colorectal cancer, we applied ridge regression on the expression profiles of proximal pathways against the 5FU $IC_{50}$ values using the colorectal organoid data.

Specifically, we used threefold cross-validation to optimize the hyperparameter for the regression model (see "Methods"). We found that "activation of BH3-only proteins" displayed high predictive performance for 5FU treatment in colorectal cancer (Fig. 2b, c). Consistent with our result, previous studies found that loss of BH3-only protein expression was associated with 5FU

**Fig. 1 Identification of biomarkers associated with drug response using a network-based machine-learning approach. a** Overall framework for the in silico identification of drug-response biomarkers through network-based machine-learning (ML). Input biological pathways for ML were first filtered to those proximal to drug targets in a protein–protein interaction network (green). The proximal pathways were then used as inputs to train the ML model (blue), revealing the predictive performance for each input pathway. Pathways with high predictive performance (orange) were selected as biomarkers and used to classify patients into drug responders and non-responders. **b** (1) Network average shortest-path lengths between drug targets and pathways were calculated to identify the proximal pathways of drug targets (see "Methods"). (2–3) Pathway expression levels for proximal pathways and drug $IC_{50}$ values from pharmacogenomic data derived from organoid models were used to train the ML model. Predictive performance was determined from the regression coefficients of the ML output, and pathways with high predictive performance were selected as drug biomarkers. (4) Expression profiles of the drug biomarkers were used to classify patients into responders and non-responders. Drug-response predictions were validated by comparing the overall survival of drug-treated patients.

treatment in many types of cancer[24–27]. We performed our analysis against other ML algorithms, such as linear and support vector regression (Supplementary Figs. 2 and 3), and found that the identified biomarker consistently displayed high predictive performance across all the algorithms, suggesting that our result did not depend on a particular algorithm. Thus, the pathway was selected as the response biomarker for 5FU efficacy.

We also applied our ML pipeline to identify drug-response biomarkers in bladder cancer after cisplatin treatment. We found 30 pathways that were proximal to cisplatin targets within the PPI network (Fig. 2e). Using bladder cancer organoid data, we measured the predictive performance of proximal pathways against the drug response of cisplatin. The "amino acid synthesis and interconversion" pathway showed high predictive performance among the proximal pathways that were consistently identified across multiple ML algorithms (Fig. 2f, g; Supplementary Figs. 2 and 3), and it was therefore selected as the biomarker for cisplatin response in bladder cancer. Consistent with our prediction, a recent study found that changes in the expression of genes associated with amino acid metabolism were attributed to cisplatin resistance in bladder cancer[28].

**Drug-response prediction in cancer patients using a network-based machine-learning approach.** We tested whether the identified drug biomarker could predict the drug-response outcomes of colorectal cancer patients after 5FU treatment. We used the biomarker transcriptomic data and clinical survival outcomes of 114 colorectal cancer patients treated with 5FU to predict and validate our approach. Statistical difference in overall survival between the predicted responders and non-responders was used as a proxy for drug-response prediction performance (Kaplan–Meier log-rank test). We observed that the predicted responders had a significantly longer overall survival compared to the non-responders (Fig. 2d; $P = 0.014$). Specifically, the responders group exhibited no deaths among the 5FU-treated colorectal cancer patients, whereas the non-responders group exhibited a 5-year overall survival of only ~50%. As a negative control, we compared the differences in overall survival of 298 colorectal cancer patients with no prior treatment who were predicted to be sensitive or resistant to 5FU treatment (Fig. 2d). From the negative control cohort, we observed no significant differences between the two classified groups ($P = 0.16$). These results suggest that the identified biomarker is predictive of drug response and is not simply associated with patient overall survival.

We also validated our approach using the data from 77 bladder cancer patients treated with cisplatin and found that the predicted responders had better survival outcomes compared to those of the predicted non-responders (Fig. 2h; $P = 0.01$). The 5-year survival rates were ~75 and 50% for the predicted responders and non-responders, respectively. Next, we investigated 294 bladder cancer patients without any previous pharmacologic intervention as a negative control cohort to test

whether our biomarker was prognostic of patient survival. We found that the expression levels of our biomarker lacked association with overall survival in the negative control cohort (Fig. 2h; $P = 0.066$). Altogether, these findings suggest that the biomarkers detected using our framework can be used to select subgroups of patients who may benefit from anti-cancer treatments, which may also dramatically reduce the use of the treatment for potential drug non-responders.

We observed that the identified biomarkers were specifically predictive of patient survival for each corresponding cancer type. To test this, we intercrossed the biomarkers found from the colorectal and bladder cancer organoids and measured their predictive ability. Specifically, we used "amino acid synthesis and interconversion" pathway to predict the drug responses of 5FU-treated colorectal cancer patients and "activation of BH3-only proteins" pathway for cisplatin-treated bladder cancer patients. The prediction performances were statistically insignificant, with log-rank $P$-values of 0.16 in the 5FU-treated colorectal cancer patients and 0.86 in the cisplatin-treated bladder cancer patients (Supplementary Fig. 4). This result suggests that the identified biomarkers are specific to the corresponding cancer and drug types.

Next, we wanted to know if the drug responders and non-responders that were classified using our method had distinguishing clinicopathologic characteristics. Therefore, we investigated whether the classified drug-response groups correlated with any clinicopathologic features. We collected age, gender, clinical tumor stage, and lymph node metastasis status data for the colorectal and bladder cancer patients who were treated with 5FU and cisplatin, respectively, and compared them with the data of the predicted drug responders and non-responders. In both cohorts, no clinical factors were significantly associated with any classified group (Supplementary Tables 1, 2). These results suggest that the expression profiles of our biomarkers are independent of other clinical features.

Recently, Aguirre-Plans et al.[29] showed that the ability of a drug to target multiple biological pathways could be leveraged to find disease-specific treatments. Thus, we wanted to know if using multiple pathways, rather than a single pathway, in our method could improve the predictive performance for drug response. We calculated the weighted sum of the expression levels of multiple pathways for individual cancer patients, using the predictive performances of the pathways in the organoid models to determine the weights (see "Methods"). We found that when using the top two pathways with the highest predictive performances, the prediction of drug responses was maintained for the cisplatin-treated bladder cancer patients (log-rank test; $P = 0.022$; Supplementary Fig. 5a), but no predictive advantage was observed for the 5FU-treated colorectal cancer patients ($P = 0.88$; Supplementary Fig. 5b). This suggests that the treatment capacities to target multiple pathways may differ across drugs and cancer types. Furthermore, the results when using up to the top 10 predictive pathways are provided in the Source data.

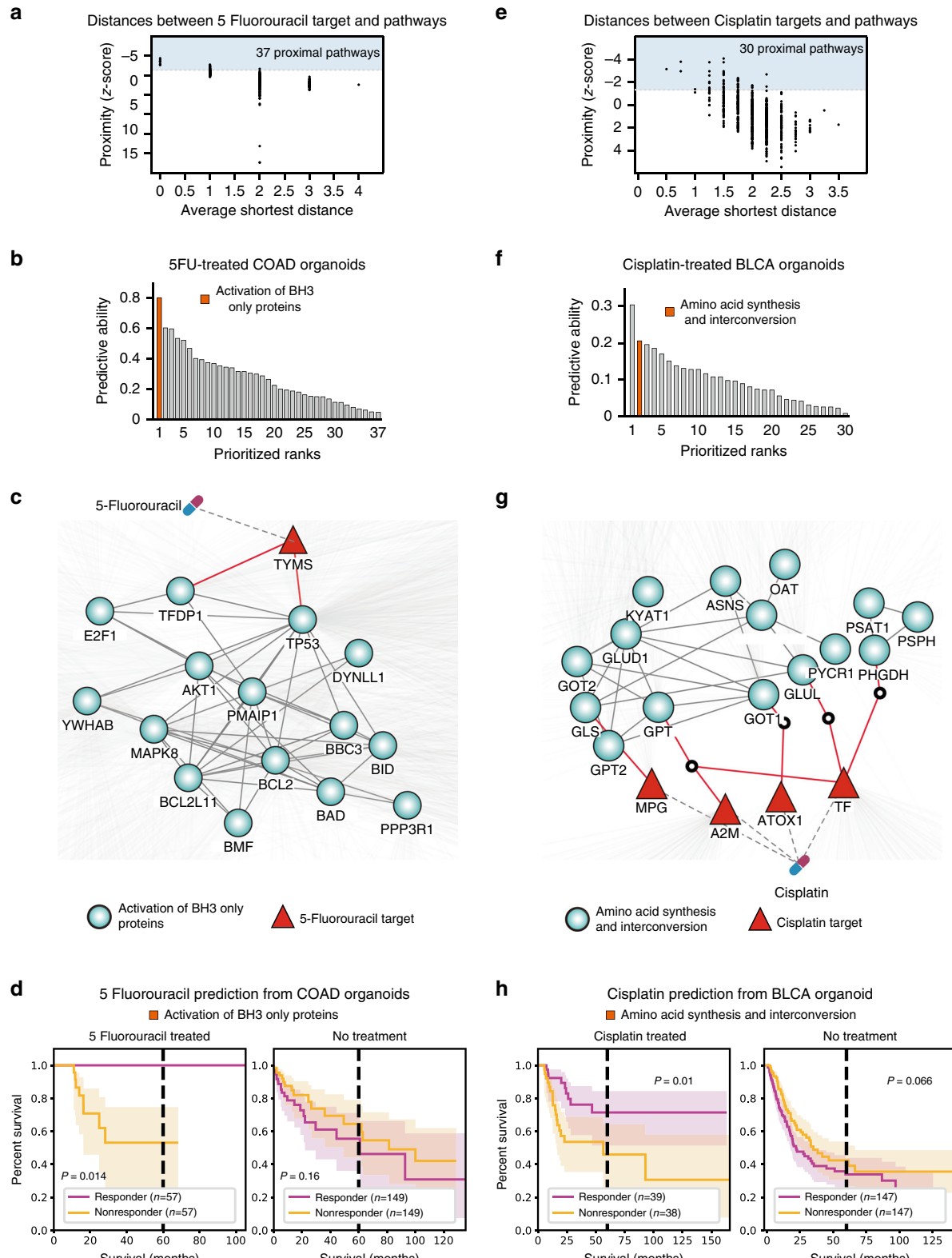

**Predictive ability of no feature selection models, alternative feature selection models, and a deep-learning model.** To demonstrate the utility of our method, we first tested the performance of drug-response prediction without a feature selection process (Fig. 3a–c). We found that the application of ridge regression models on whole-genome or whole-pathway transcriptomic data from 3D organoid models was not predictive of

drug responses in both 5FU-treated colorectal and cisplatin-treated bladder cancer patients (Fig. 3b, c). This result suggests that feature reduction may be required prior to training an ML model for drug-response prediction.

Next, we replaced the network proximity-based feature selection process with other feature selection procedures to compare drug-response prediction performances. We tested

**Fig. 2 Identification of biomarkers in organoid models and validation in human tumors. a** Network distances and z-scores of Reactome pathways for 5-fluorouracil. Proximal pathways (see "Methods") are shown in blue. **b** Predictive performances of pathways from colorectal cancer organoid models. Ridge regression was performed to calculate predictive performances. Selected drug-response biomarker for clinical drug-response prediction is shown in orange. **c** Network representation of 5-fluorouacil target and the biomarker pathway. **d** (Left) Drug-response predictions for 5-fluorouracil-treated colorectal cancer (COAD). Predicted responders and non-responders are depicted in purple and yellow, respectively. (Right) Predictions for patients with no known treatment history are shown for COAD patients. Number of patients is shown inside the parenthesis. 95% confidence interval was used. Statistical significance was measured using Kaplan–Meier survival curves and two-sided log-rank tests. P-values <0.05 were considered significant. **e, f** Biomarker identification from cisplatin-treated bladder cancer (BLCA) organoids. **g** Network representation of cisplatin targets and the biomarker pathway. **h** Drug-response prediction for cisplatin-treated BLCA patients. Kaplan–Meier survival plots and two-sided log-rank P-values are displayed.

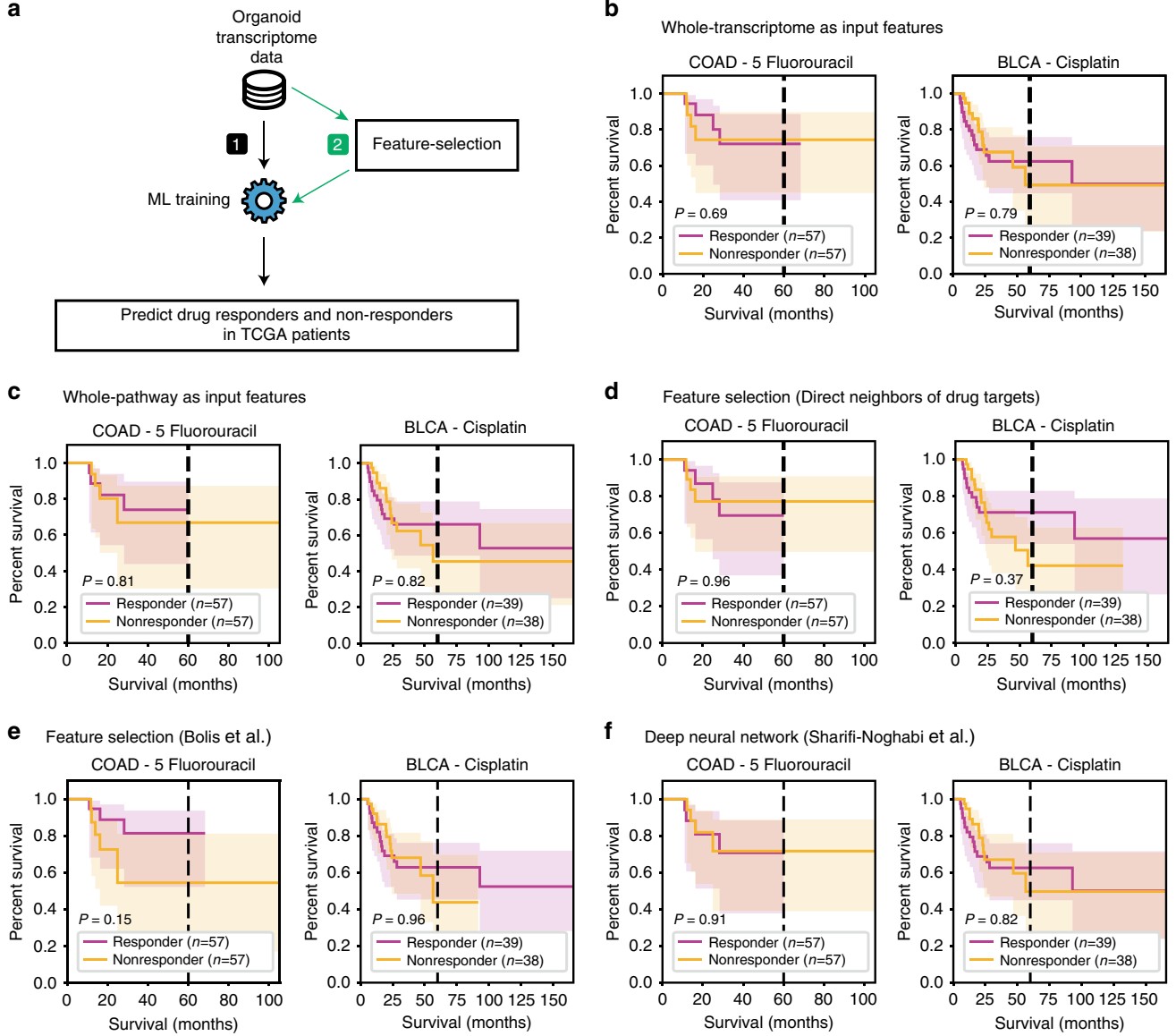

**Fig. 3 Drug-response predictions using other machine-learning methods. a** Overall scheme of other drug-response prediction methods using (1) no feature selection and (2) a feature selection process. **b**–**d** Survival predictions were trained using (**b**) whole-genome profiles, (**c**) whole-pathway profiles, (**d**) first neighbors of drug targets, (**e**) feature selection (Bolis et al.), and (**f**) a deep-learning method (Sharifi-Noghabi et al.). Kaplan–Meier survival plots and two-sided log-rank test P-values are displayed. COAD, colorectal cancer; BLCA, bladder cancer.

predictive performances of feature selection based on (1) network centrality, (2) direct network neighbors of drug targets, and (3) the pre-selection of genes with expression levels highly correlated to drug responses (see "Methods"). We found that all of the tested feature selection methods were not strong predictors of patient survival (Fig. 3d, e, Supplementary Figs. 6 and 7). More specifically, for feature selection based on network centrality, we

used three different measurements (degree, betweenness, and closeness centrality) to calculate the centrality of each pathway (see "Methods" for details). Feature selection was conducted by selection of network central pathways that matched the number of proximal pathways. We found that all of the combinations of different feature selection procedures based on network centrality were not strongly predictive of patient survival after drug

treatment (Supplementary Figs. 6 and 7). Furthermore, we found that the expression levels of drug targets or up to the first, second, or third-degree network neighbors failed to classify patient survival with statistical significance (Fig. 3d, Supplementary Figs. 8 and 9). These results supported recent observations that the expression profiles of drug targets alone as well as the network neighbors of drug targets were not predictive of drug response[6,7,30,31], suggesting that utilization of network proximity allowed robust signals to be captured at the pathway level, but not at the gene level. Moreover, we also applied the feature selection procedure by Bolis et al., which leverages 10 iterations of leave-half-out cross-validation to select genes for ML training (see "Methods"). The trained ML models showed less predictive power, with log-rank test $P$-values of 0.15 in 5FU-treated colorectal and 0.96 in cisplatin-treated bladder cancer patients (Fig. 3e). These results provide further evidence that feature selection based on pathway-level network proximity is useful to find robust biomarkers.

Finally, we compared our method to a deep-learning method developed by Sharifi-Noghabi et al.[32]. We observed that the deep-learning model was not predictive of patient survival in 5FU-treated colorectal and cisplatin-treated bladder cancer patients. Briefly, a combined loss function using a triplet loss and a binary cross-entropy loss was implemented to optimize the prediction model (see "Methods"). The prediction performances were measured by the log-rank test, with $P$-values of 0.91 for the 5FU-treated colorectal and 0.82 for the cisplatin-treated bladder cancer patients (Fig. 3f). Of note, the deep-learning method was developed for multi-omic purposes; therefore, the prediction performance may have declined when it was trained with transcriptomic data alone.

**Validation of identified drug-response biomarkers in drug-sensitive and -resistant isogenic cancer cell lines.** Motivated by the predictions of the network-assisted ML algorithm, we next tested whether the molecular profiles of identified drug bio-markers were mechanistically associated with drug sensitivity. We observed transcriptomic biomarker rewiring in 5FU-sensitive and -resistant isogenic colorectal cancer cell lines (Fig. 4a, b). Specifically, we examined a dataset measuring the gene expression profiles of drug-sensitive and -resistant isogenic colorectal cancer cell lines, in which resistance was elicited via exposure to gradually increasing concentrations of 5FU[33] (Fig. 4a). We hypothesized that the colorectal cancer cell transcriptomic signatures associated with drug response would change following the development of resistance to 5FU. When measuring the expression levels of the "activation of BH3-only proteins" pathway in the 5FU-sensitive and -resistant colorectal cancer cell lines, we observed significant changes in the biomarker pathway expression levels between the two types of cells (Fig. 4b). Specifically, we found that the 5FU-sensitive cells overexpressed components of the biomarker pathway compared to the 5FU-resistant cells (two-sample, two-tailed Student's $t$-test; $P = 0.0085$), which was consistent with our model's predictions.

We also evaluated any changes in the transcriptomic signatures of cisplatin-sensitive and -resistant isogenic bladder cancer cell lines and found alterations in the expression levels of the identified biomarker pathway between the two types of cells (Fig. 4c). Specifically, we investigated the "amino acid synthesis and interconversion" pathway, which was selected as the drug-response biomarker for the cisplatin-treated bladder cancer organoids. To investigate the potential role of the biomarkers in drug sensitivity, we measured the transcriptomic profiles of the isogenic cisplatin-sensitive and -resistant bladder cancer cell lines. Because biomarker pathway overexpression was associated with

increased drug sensitivity in the network-based ML model, we hypothesized that cisplatin-resistant cells would demonstrate reduced expression of the biomarker pathway. Accordingly, we observed significantly lower expression levels of the biomarker pathway in the cisplatin-resistant cells compared to the cisplatin-sensitive cells (Fig. 4c; $P = 0.00022$). Similar to our data, previous studies reported that amino acid metabolism was mechanistically associated with cisplatin sensitivity in bladder cancer[28]. Altogether, these findings suggest that our ML model provides a mechanistic hypothesis for drug responses that can be tested against experimental analysis. Expression differences between drug-sensitive and -resistant cancer cell lines for the top three proximal pathways are provided in Supplementary Table 3.

To confirm that the identification of predictive biomarkers was not obtained by arbitrary feature selection, we performed a bootstrapping analysis that aimed to determine whether randomly selected features could predict the drug responses of 5FU-treated colorectal and cisplatin-treated bladder cancer cells. For 10,000 iterations, we first selected random pathways that matched the number of proximal pathways for each anti-cancer drug. Next, we conducted ridge regression using the transcriptomes of randomly selected pathways and the drug-response data from organoid models. Finally, in each iteration, we ranked pathways that were predictive of patient survival and drug sensitivity in isogenic cancer cell lines (Fig. 4d). We observed that our predictions using proximal pathways were statistically significant, as measured by empirical $P$-values (0.0012 and 0.014 in 5FU-treated colorectal cancer and cisplatin-treated bladder cancer, respectively), and were not likely to be observed by random feature selection (Fig. 4e, f). In addition, we found that the number of genes in a pathway was not a strong predictor of drug response in cancer patients, suggesting that it was not a confounding factor in our analysis (Supplementary Fig. 10). All of the predictive pathways that were ranked equal to or higher than our biomarkers are presented in Supplementary Table 4. Taken together, these results imply that the combined use of gene modules and network analysis in organoid pharmacogenomic models may be required to identify robust biomarkers for the prediction of drug response.

**Identification of known drug-response biomarkers using a network-based machine-learning approach.** To further validate our network-based ML approach, we investigated whether our method could identify known clinical drug-response biomarkers. We compared our predictions with known drug-response bio-markers derived from independent datasets, according to previous work by Geeleher et al.[34] and Mourragui et al.[35] (see "Methods"). Specifically, we calculated the predicted drug resistance scores for each patient using our network-based approach using transcriptomic data, and then we investigated whether the predicted drug resistance scores correlated with alterations of known drug-response biomarkers (i.e., mutation status).

First, we investigated the presence of a BRAF$^{V600E}$ mutation, which is known to be reliably predictive of resistance to the EGFR inhibitor cetuximab in colorectal cancer. Using our network-based approach, we found that the "Gastrin-CREB signaling pathway via PKC and MAPK" pathway was associated with the response to cetuximab treatment in colorectal cancer organoids (Supplementary Fig. 11). Thus, we used the expression levels of the predictive pathway components to derive the predicted drug resistance scores (see "Methods"). We found that the predicted drug resistance scores were significantly higher in patients bearing the BRAF$^{V600E}$ mutation compared to those with wild-type BRAF (Fig. 5a; one-sided Mann–Whitney U test; $P = 0.037$). This result was consistent with previous observations in which

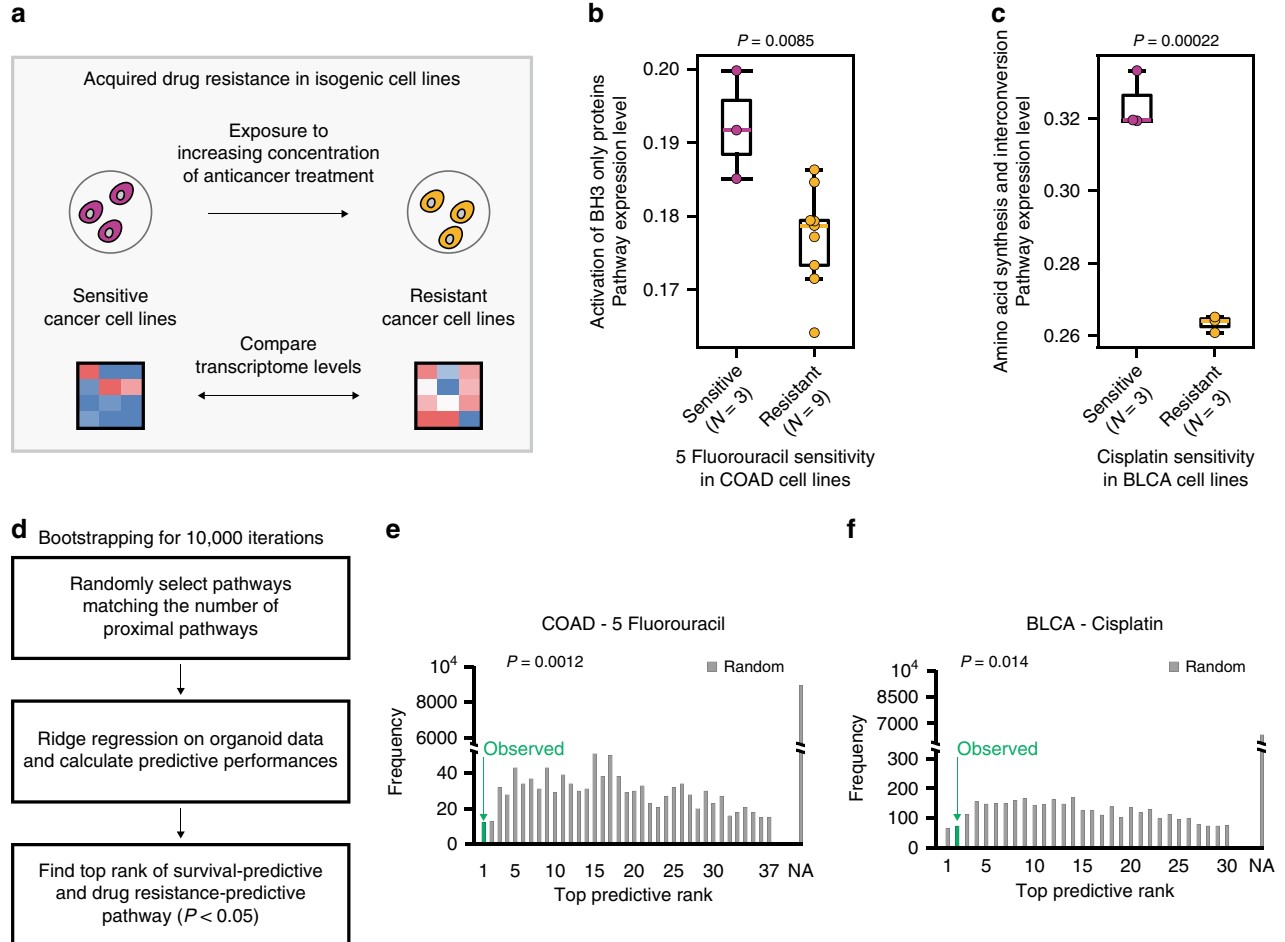

**Fig. 4 Validation of drug-response biomarkers in independent datasets. a** Overall scheme of the analysis. Isogenic drug-sensitive and -resistant cell lines were used for the analysis. Transcriptomic differences between the drug-sensitive and -resistant cell lines were investigated. **b** Differences in the expression levels of the "activation of BH3-only proteins" pathway between 5-fluorouracil-sensitive ($n = 3$) and -resistant ($n = 9$) colorectal cancer cell lines. Single sample GSEA (ssGSEA) was performed to calculate pathway expression levels. A two-sample, two-tailed Student's t-test was performed to measure statistical significance. Boxplot displays median value, interquartile range (IQR) as bounds of box and whiskers extending to upper/lower quartile ± IQR × 1.5. **c** Differences in the expression levels of the "amino acid synthesis and interconversion" pathway in cisplatin-sensitive ($n = 3$) and -resistant ($n = 3$) cell lines. A two-sample, two-tailed Student's t-test was performed to measure statistical significance. Boxplot displays median value, IQR as bounds of box and whiskers extending to upper/lower quartile ± IQR × 1.5. **d** Overall scheme of drug prediction from bootstrapping analysis. **e, f** Statistical significance of biomarker rank for colorectal cancer after 5-fluorouracil treatment (**e**) and bladder cancer after cisplatin treatment (**f**). Statistical significance was computed using empirical P-value, which was calculated by the frequency of iterations with the top predictive ranks that were equal to or higher than the observed ranks divided by the total number of iterations (10,000). Random bootstrapping results are shown in gray. The observed rank from using proximal pathways to train ridge regression is displayed in green. If no pathway was predictive of survival for the random pathway sets, NA (not available) was provided. COAD, colorectal cancer; BLCA, bladder cancer.

colorectal adenocarcinoma patients with BRAF[V6600E] mutation were resistant to cetuximab treatment[36,37]. We compared our results from the network-based method with other ML methods used in Fig. 3 (whole-transcriptome, whole-pathway, or drug target neighbors). Except for our network-based method, none of the other comparable methods were able to identify the association between BRAF[V600E] mutation and cetuximab resistance in colorectal cancer (Supplementary Fig. 12). These results further demonstrate the utility of a network-based approach to discover robust biomarkers in cancer.

Next, we tested the association between mutations in the excision repair cross-complementing 2 (ERCC2) DNA repair gene and the cisplatin drug response in bladder cancer. We used the expression profiles of the "amino acid synthesis and interconversion" pathway to measure the predicted drug resistance scores for bladder cancer patients. We observed that the presence of ERCC2 mutations predicted lower drug resistance

to cisplatin (Fig. 5b; $P = 0.002$), which was also observed in previous reports in which ERCC2 mutations sensitized patients to cisplatin treatment[38,39]. All of the drug-response predictions using other ML methods showed reduced statistical power compared to our method (Supplementary Fig. 13). Altogether, these results suggest that a network-based approach is useful to capture the concordance across omic datasets for prediction of drug sensitivity.

## Discussion
In this study, we tested if the incorporation of network analysis into an ML framework could accurately identify robust drug-response biomarkers using organoid models. Indeed, we found that our method accurately predicted cancer patient–drug responses, whereas conventional ML approaches showed less optimal predictive performances. Importantly, our network-based ML model provided interpretable results for drug-response

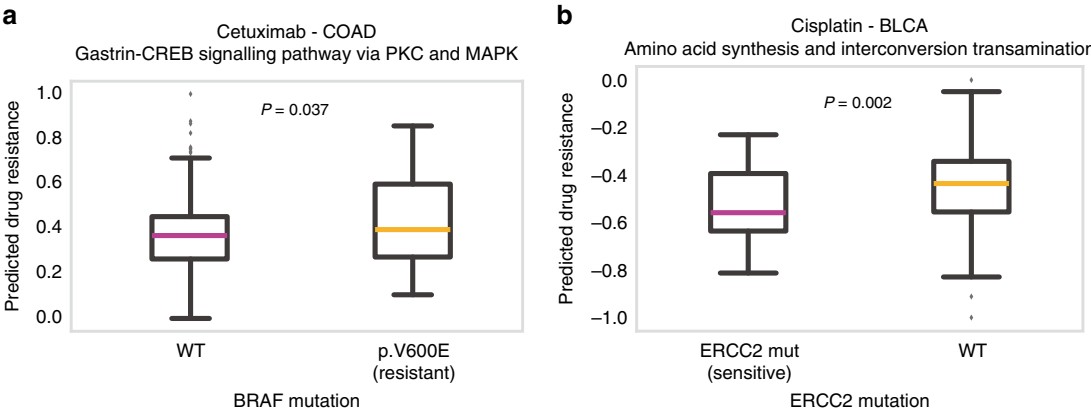

**Fig. 5 Association of predicted biomarkers with known biomarkers from independent molecular datasets. a** Predicted drug responses (IC$_{50}$ values; see "Methods") for colorectal tumors based on cetuximab responses that were trained from colorectal organoid models, using only transcriptomic data. The "Gastrin-CREB signaling pathway via PKC and MAPK" pathway was used to measure predicted drug resistance. Predicted drug resistance of 328 BRAF wild-type and 47 BRAF$^{V600E}$ mutated colorectal cancer (COAD) patients are displayed. The BRAF$^{V600E}$ mutation status correlated with predicted drug resistance. Statistical significance was measured by a one-sided Mann–Whitney U test. **b** Predicted drug resistance in cisplatin-treated bladder cancer (BLCA) patients. Predicted drug resistance was calculated using the "amino acid synthesis and interconversion" pathway. Predicted drug resistance of 367 ERCC2 wild-type and 37 ERCC2-mutated bladder cancer patients are displayed. Boxplot displays median value, interquartile range (IQR) as bounds of box and whiskers extending to upper/lower quartile ± IQR × 1.5. Statistical significance was measured by a one-sided Mann–Whitney U test.

prediction, which were further tested in external experimental datasets. Our results suggest that network-based ML in organoid models can improve drug-response prediction for cancer patients.

Interpretability of ML models is becoming increasingly important, especially when it comes to the high-stakes decision-making of healthcare[40]. To generate interpretable ML models, incorporation of biological structures, such as PPI networks, into the ML models assists in making comprehensive predictions[10,11]. Recently, Ma et al. showed that the infusion of hierarchical structures of biological functions into neural networks accurately predicted cellular growth from genotypes[41]. Unlike standard deep-learning methods, interpretable prediction results provide information on the underlying biological mechanisms of genotype–phenotype relationships[10,41] which can be experimentally validated and leveraged for the development of novel, personalized, targeted therapies or for an increased understanding of a disease.

Our research further supports the utility of network analysis to reduce the biological complexity of input datasets by capturing robust signals, which can improve the predictive performances of ML tasks[10,11]. Recently, the PsycheENCODE Consortium embedded a regulatory network into a deep-learning model and demonstrated that functional genomic data improved the detection of gene-phenotype associations in neurological disorders[42]. In addition, Yang et al. used an ML approach that learned via biologically known cause-and-effect relationships by utilizing metabolic flux networks, which revealed interpretable treatment effects of antibiotics[43]. Moreover, Hofree et al. found that subtyping cancer patients into groups based on the harboring of mutations in a similar region of a PPI network was predictive of overall survival[44]. In addition, we previously reported that PPI networks could be leveraged to identify various genotype–phenotype associations, including disease–gene relationships[45–49], the clinical severity of human diseases[50], gene essentiallity[51,52], and phenotypic outcomes of chemical treatments[53]. Altogether, network analysis may help detect effective biological signals that can be manipulated for ML tasks, such as drug-response prediction.

Currently, 5FU is the standard treatment for colorectal cancer; however, only 10–15% of patients have a measurable response to this therapy[24]. Thus, there is an urgent need to develop computational methods for the discovery of novel biomarkers for 5FU resistance in colorectal cancer. Our method identified such biomarkers (Fig. 2). We found that the expression levels of the components of the "activation of BH3-only proteins" pathway were associated with the responses of colorectal cancer organoids to 5FU treatment. Because BH3-only proteins are essential initiators of apoptosis in response to intracellular damage[26,54], their activity levels may affect anti-cancer drug responses that induce DNA damage. Thus, the activity of BH3-only proteins is likely associated with 5FU drug sensitivity because the drug works by (1) blocking thymidylate synthase and inhibiting DNA replication and repair and by (2) greatly impacting cell metabolism and vitality via its incorporation into DNA and RNA[24]. Indeed, previous studies have demonstrated that 5FU resistance is attributed to the loss or reduction of BH3-only protein expression levels in various cancer types[24–27], which was consistent with our findings. These results suggest that there is a connection between BH3-only protein activity and 5FU sensitivity; however, this relationship requires further elucidation in colorectal cancer.

We also discovered that our model could predict the therapeutic response to cisplatin in bladder cancer. Cisplatin is commonly administered to bladder cancer patients, but only 30–40% of the patients exhibit a response with this therapy[55], suggesting that predictive biomarkers are essential to drastically improve the benefit of the treatment. We found that the elevated expression of the components of the "amino acid synthesis and interconversion" pathway was associated with cisplatin response in bladder cancer organoids. This association was further validated by survival prediction in bladder cancer patients after cisplatin treatment (Fig. 2). These data were consistent with a recent report in which amino acid and polyamine metabolism-associated gene promoters were hypermethylated, resulting in the lower expression of these genes in cisplatin-resistant bladder cancer cell lines[28]. Because amino acid metabolism is closely related to cancer progression[56,57], several clinical trials are currently being conducted (e.g., ADI-PEG 20 treatment) that are tailored toward depleting the amino acid supply to cancer cells that are highly dependent on amino acid metabolism[58,59]. Although it would be interesting to investigate whether amino acid depletion therapy improves cisplatin efficacy in bladder

cancer, the mechanistic role of amino acid metabolism in bladder cancer resistance requires further clarification.

We were able to validate our biomarker analysis framework using external molecular measurements in drug-sensitive and -resistant isogenic cancer cell lines (Fig. 4). Although no information on the molecular states before and after the drug treatments was provided for biomarker discovery, our molecular biomarker profiles were altered in concordance with our model predictions from the organoid data. In the future, paired datasets that provide molecular changes prior to and after drug treatments, such as the Library of Integrated Network-based Cellular Signatures (LINCS) L1000 datasets[60], could be leveraged into the ML model to improve the prediction of drug responses or identification of biomarkers. Also, Acar et al. recently reported a procedure that generated a large population of reproducible, drug-resistant cancer cells[61], which were used to further validate drug-response biomarkers. However, one caveat of using cancer cell lines is that some cancer cells do not accurately represent primary human tumors[17,62]. Accordingly, compared to using patient-derived organoid data, we found that using large-scale pharmacogenomic data of all the cancer cell lines did not strongly predict survival in 5FU-treated colorectal and cisplatin-treated bladder cancer patients (Supplementary Fig. 15). This suggests that careful selection of cancer cell lines with high resemblance to primary tumors is necessary for successful translational medicine[17,62,63].

Because concordance between multiple molecular levels for the prediction of drug sensitivity was previously shown[64], we wanted to know whether our biomarker predictions were concordant with other known biomarkers derived from non-transcriptomic data. Indeed, we observed that our biomarkers identified from transcriptomic data correlated with the known biomarkers detected from somatic variants, further supporting the existence of concordance between molecular layers (Fig. 5). Consistent with our results, a recent study integrated multi-omic data of cancer cell lines into network modules to improve the discovery of biomarkers and the prediction of therapeutic responses[5]. Although an ML model that integrates various molecular data types could better facilitate the discovery of robust therapeutic biomarkers from organoid models, such analyses would require further molecular layer profiling in organoids, which is beyond the scope of this study.

Because clinical trials require vast resources, we envision that the translation of predictive biomarkers identified in preclinical models to human tumors will continue to be an active area of research in cancer pharmacogenomics. Importantly, we expect that this translation will become increasingly important due to the high molecular similarities between the organoid models and human tumors as well as the capacity of the organoid models to reflect drug treatment outcomes that are comparable to those observed in human clinical trials[17,18]. As organoid models become more sophisticated with the incorporation of additional microenvironment and immune components, we believe that biomarker discovery for cancer therapies will more quickly advance.

## Methods

**Organoid model and human tumor pharmacogenomic data**. We collected gene expression and drug-response data of colorectal and bladder cancer organoids from van de Wetering et al.[22] and Lee et al.[23], respectively. Drug IC$_{50}$ values were used for drug sensitivity measurements. For microarray data, we used robust multi-array average (RMA)[65] normalized expression data. For cancer patient data, we curated mutation and gene expression, drug treatment status, and clinical outcomes from TCGA data (https://www.cancer.gov/tcga) for colorectal (TCGA-COAD) and bladder (TCGA-BLCA) cancers. Sequencing of human participants was performed by TCGA consortium under a series of locally approved Institutional Review Board (IRB) protocols as described by the consortium[66]. We downloaded the FPKM-UQ

(upper quartile) dataset from TCGA data portal for expression analysis. The patient expression values were further converted to log$_2$ (FPKM-UQ + 1) for convenient analysis. Because various synonymous drug identifiers were used in TCGA clinical drug data, drug names were standardized using the DrugBank IDs (https://go.drugbank.com/) to provide consistent naming between the two data sources[67]. All gene IDs were mapped to Uniprot IDs (https://www.uniprot.org/)[68]. To estimate expression activity levels of each pathway, we performed single sample GSEA (ssGSEA) analysis using the GSEAPY python module, and the normalized enrichment score (NES) was used to indicate pathway activation levels[69]. We performed z-score standardization on each gene or pathway across all samples within a dataset to ensure a mean equal to zero and standard deviation of one. Finally, we assessed batch effects between the organoid and patient data using principal component analysis (PCA) (Supplementary Fig. 1).

**Preparation of protein–protein interaction network, drug target, and pathway data**. We derived the human PPI network from the STRING database (https://string-db.org/)[21]. For high-confidence links, we used interactions with confidence scores >700, as done by Fernández-Torras et al.[7]. We used the largest connected component of the interactome for our analysis, which resulted in 323,774 interactions between 13,824 proteins. Uniprot Gene IDs were used to map genes to the corresponding proteins in the interactome.

To calculate the proximity of drugs to pathways, we downloaded drug–drug target associations from DrugBank and the Reactome pathway from the MSigDB database (C2: REACTOME [https://www.gsea-msigdb.org/gsea/msigdb/collections.jsp])[70–72]. All genes found within pathways as well as annotated drug targets were then mapped to Uniprot IDs. Finally, the network was used to filter pathways and drugs with no genes in the PPI network.

**Network-based proximity between drugs and pathways**. The distance between the pathway genes and drug-associated genes that were normalized by the random expectation defined the proximity between pathways and drug-associated genes. The network-based distance was calculated using the closest distance parameter[14,15]. The measurement was represented by the average of the shortest-path lengths between drug-associated genes and the nearest pathway genes, according to the following equation:

$$d_c = \frac{1}{|T|} \sum_{t \in T} \min_{s \in S} d(s, t) \tag{1}$$

where $T$ is the set of drug-associated genes, $S$ represents the pathway genes, and $d(s, t)$ is the shortest path between the drug-associated and pathway genes. Codes for calculating network proximity were downloaded from https://github.com/emreg00/toolbox. All of the calculations were done using python packages including pandas (v 0.24.2), matplotlib (v 2.0.0), numpy (v 1.16.6), scipy (v 1.2.2), sklearn (v 0.20), lifelines (v 0.19.5), and gseapy (v 0.10.1).

To assess the significance of the distance $d(s,t)$, we bootstrapped random genes to produce a reference distribution. We randomly selected genes by matching the size and the number of network neighbors (degree) of the original drug-associated genes and pathway genes and computed the closest distance. We repeated the procedure for 1000 iterations, and the mean and standard deviation of the reference distribution were used to calculate a z-score. A pathway resulting in a z-score lower than 90% ($\alpha = 0.10$) of the reference distribution scores (z-score ≤ −1.2816) was considered proximal.

To test whether the randomized procedure successfully limited high-dependency toward central pathways, we checked if some pathways were always proximal to anti-cancer drugs. We tested 44 drugs that were screened in colorectal and bladder cancer organoids by van de Wetering et al.[22] and Lee et al.[23], respectively. We found that only 3 out of 674 pathways (0.45%) were proximal to more than half of the drugs tested (proximal to 22 drugs or more), whereas on average, a pathway was proximal to 4.9 drugs (median: 4 drugs per pathway), suggesting that randomized process was useful to reduce potential bias toward any network central pathways (Supplementary Fig. 16).

**Calculation of pathway predictive performance via machine-learning training**. For ML training, we used linear regression, ridge regression, and support vector regression, which were all implemented using Scikit-learn in python[73]. To train ML algorithms, we used expression profiles of genes/pathways against drug IC$_{50}$ values. Default settings of the linear-regression model (sklearn LinearRegression python module) were used to build linear regression. For support vector regression, we used linear kernel (sklearn SVR python module). Ridge regression was performed using the sklearn RidgeCV function. The optimal $\alpha$ value was selected using RidgeCV's cross-validation function (threefold cross-validation) by iterating $\alpha$ from 0.1 to 1 using a 0.1 interval. The regression model with the optimal α value was used to train organoid models. Pathways were ranked based on the magnitude of their regression coefficient, and then we defined the magnitude as the pathway predictive performance. Pathways with high predictive performance were selected as drug-response biomarkers.

**Predictive ability of proximal pathways in organoid models**. To test the performance of proximal pathways in predicting drug response for organoid models,

we split the organoid dataset into training (60%), validation (10%), and test (30%) sets, similar to what was done by Bolis et al.[74]. We used ridge regression to train and predict drug responses, and we iterated the $\alpha$ value from 0.1 to 1 using a 0.1 interval to train the training set and to predict drug response in the validation set. The optimal $\alpha$ was selected where the root-mean-squared error was lowest in the validation set. The optimized model was then used to predict the drug response in the test set. The final predictive performance was measured by comparing the correlation between the observed and predicted drug responses in the test set ($R^2$). To test whether a specific validation set could have effects on our results, we sampled all possible combinations of validation sets within the training set to predict drug responses in the test set. We found high correlation between the observed and predicted drug responses in both colorectal cancer ($R^2 = 0.98$) and bladder cancer organoids ($R^2 = 0.89$) (Supplementary Fig. 17), which suggests that training the ML model with transcriptomes of proximal pathways has predictive ability.

**Inference of patient-specific drug responses**. To infer patient-specific drug responses, we introduced a measure that takes the expression levels of a pathway and its corresponding regression coefficient, which was computed from the organoid models. The drug resistance score for each patient–drug pair was calculated according to the following function:

$$\text{Score}_{\text{patient}} = \text{Exp}_{\text{patient}} \times \beta_{\text{preclinical}} \qquad (2)$$

$\text{Exp}_{\text{Patient}}$ represents the expression levels of a pathway for a given patient and $\beta_{\text{preclinical}}$ is the regression coefficient (using either linear, ridge, or support vector regression) from preclinical models.

To enable prediction models that utilize multiple pathways to infer each patient's drug response, we first ranked pathways from the highest to lowest predictive performance. The Eq. (2) was modified into the following function:

$$\text{Score}_{\text{patient}} = \sum_{p \in P}^{N} \text{Exp}_{\text{patient},p} \times \beta_{\text{preclinical},p} \qquad (3)$$

where $N$ is the total number of the top predictive pathways selected for multi-pathway analysis, $p$ is a pathway among the selected top predictive pathways, $\text{Exp}_{\text{Patient},p}$ is the expression levels of pathway $p$ in a given patient, and $\beta_{\text{preclinical},p}$ is the regression coefficient for the pathway $p$.

Based on the inferred drug resistance, we rank-ordered patients and split them into two groups (responders vs. non-responders) using the median as a cut-off, as done by Nickerson et al.[75]. Kaplan–Meier survival analysis was conducted to visualize and calculate the significance of the differences between the predicted responders and non-responders. Kaplan–Meier survival plots were visualized using Oasis2[76,77] and the lifelines package from python[78]. Log-rank test $P$-values were calculated using the lifelines package to quantify statistical differences in overall survival[78].

**Feature selection process based on network centrality**. We separately used degree, betweenness, and closeness centrality to compute network centrality scores of each pathway. To define the representative centrality score for a pathway, we calculated the average of all of the centrality levels of genes within the pathway. For each drug, we selected features with high pathway centrality scores, while matching the number of proximal pathways identified for the drug. The selected network central pathways were tested against organoid pharmacogenomic data using ridge regression.

**Feature selection process based on a method by Bolis et al**. We used a method provided by Bolis et al.[74] to select features for drug-response prediction. We selected genes that showed significant average correlation (Spearman $P$-value <0.05) with the $IC_{50}$ values from the organoid screening data using 10 iterations of leave-half-out cross-validation, as provided by the authors. These genes were then used to train the ridge regression model and to predict drug response in cancer patients. The median of the predicted $IC_{50}$ values was then used to classify the cancer patients as drug responders or non-responders. Of note, we were unable to implement filtering genes by removing genes that were missing in the co-expression network, as done by the authors, because no gene was retained after co-expression network comparison using the organoid datasets.

**Deep-learning-based prediction using a method by Sharifi-Noghabi et al**. We applied a deep-learning method by Sharifi-Noghabi et al.'s[32] to our data using transcriptomic organoid and patient data. To train organoid transcriptomic data, we divided the organoid samples into drug responders and non-responders based on the median $IC_{50}$ value. We performed cross-validation to tune the hyperparameters of the deep neural networks. The optimal hyperparameters were selected by maximizing the area under the receiver-operating characteristic curve (AUC). The tuned model trained on the organoid data was then used to predict the response category for patients. All tested and selected hyperparameters are provided in the Source data.

**Analysis of isogenic drug-sensitive and -resistant cell lines**. Microarray transcriptomic data of 5FU-sensitive and -resistant isogenic colorectal cancer cell lines were downloaded from the Gene Expression Omnibus (GEO) database under the accession number GSE81008[33]. Transcriptomic data of cisplatin-sensitive and -resistant isogenic bladder cancer cell lines were obtained from Yeon et al.[28] upon personal request. Gene expression profiles were transferred to pathway expression levels using the single sample GSEA (ssGSEA) tool. Unpaired, two-tailed, two-sample Student's $t$-tests were used to quantify statistical differences in the pathway expression levels between the drug-sensitive and -resistant cell lines.

**Association between known drug-response biomarkers and expression levels**. Following the previous work by Geeleher et al.[34] and Mourragui et al.[35], we compared the predicted drug responses using equation (1) to the mutational statuses of known drug responses using TCGA cancer patient data. We investigated the effect of $BRAF^{V600E}$ mutation on cetuximab (EGFR inhibitor) resistance in colorectal cancer and of $ERCC2$ mutations on cisplatin resistance in bladder cancer.

To analyze cetuximab resistance in colorectal cancer, we tested the extent to which organoid samples could be removed and the associations with the mutational biomarkers could still be recovered. We removed up to three organoid samples to train the ML models and their performances to recover drug responses based on known mutational biomarkers. Performances were measured by one-sided Mann–Whitney U tests ($P$-value <0.05 was considered significant). We computed the robustness of our approach by calculating the fraction of organoid sets that recovered drug responses, using all of the possible combinations of organoid samples to train ML models. We found that over 70% of the organoid sets recovered drug responses when three samples were removed from the original data (Supplementary Fig. 18). As a negative control, we trained an ML model using whole-transcriptome or whole-pathway models and observed reduced robustness compared to our method, demonstrating the robustness of our network-based approach (Supplementary Fig. 18). We also showed the prediction results when using reduced sample sizes, in which three organoid samples with hypermutations (P19a, P19b, P24a) were removed from the analysis to reduce potential confounding factors (Fig. 5a and Supplementary Fig. 12).

**Pharmacogenomic data of cancer cell lines**. We collected pharmacogenomic data from the Genomics of Drug Sensitivity in Cancer (GDSC) database[79]. We separately tested tissue type-specific cancer cell lines and all available cancer cell lines to train the ML models because previous reports found that utilizing all cancer cell lines compared to tissue type-specific cancer cell lines improved the prediction performance of patient–drug responses for some anti-cancer drugs[1]. Potential batch effect was corrected via z-score standardization, and the reduction of batch effect was visualized by a PCA plot (Supplementary Fig. 14).

## Data availability
Gene expression and drug-response data of colorectal cancer (COAD) organoids were downloaded from the Gene Expression Omnibus (GEO) with the accession code GSE64392 and from supplementary material from Wetering, Marc et al.[22]. Pharmacogenomic data of bladder cancer (BLCA) organoids were downloaded from supplementary material from Lee et al.[23]. Gene expression, mutation, drug treatment status and clinical outcomes of COAD and BLCA cancer patients were downloaded from the TCGA portal (https://www.cancer.gov/tcga). 5-Fluorouracil-sensitive and -resistant COAD isogenic cell lines were downloaded from GEO with the accession code GSE81008[33]. Large pharmacogenomic data of cancer cell lines were downloaded from GDSC data portal (https://www.cancerrxgene.org/)[79]. STRING protein–protein interaction network was downloaded from the STRING website (https://string-db.org/)[21]. Drug and drug target relationship was downloaded from drugbank portal (https://go.drugbank.com/). Gene IDs mapped to Uniprot ID were downloaded from UNIPROT portal (https://www.uniprot.org/)[68]. REACTOME pathway data was downloaded from MSigDB database (C2: REACTOME [https://www.gsea-msigdb.org/gsea/msigdb/collections.jsp])[70–72]. Source data are provided with this paper.

## Code availability
The source codes for reproduction of the results were developed in python 2.7 and are available at a GitHub repository (https://github.com/billy-kong/organoid_biomarker_detection).

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

## Acknowledgements

We thank all members of the Kim and Shin laboratory for the helpful discussions. This work was partially supported by grants from the Korean National Research Foundation (2018R1A2B6002657, 2017M3C9A6047625, and 2020R1A6A1A03047902) and a grant from IITP (2019-0-01906, Artificial Intelligence Graduate School Program, POSTECH).

## Author contributions

J.K., H.L., K.S., and S.K. conceived and designed the experiments. J.K. performed the experiments. J.K., H.L., D.K., S.H., and D.H. analyzed the data. J.K., H.L., D.K., S.H., D.H., K.S., and S.K. wrote the manuscript.

## Competing interests

The authors declare no competing interests.
