## [Peer Review File · Nature Communications]

Reviewers' Comments:

Reviewer #1:

Remarks to the Author:

The authors present an approach to identify transcriptional signatures predictive of drug response via a network based analysis of public available pharmacogenomic data, from the characterisation of biobanks of colorectal and bladder cancer organoids.

To this aim, the authors use a statistical strategy to identify pathway gene sets that are more proximal than expected by random chance to the nominal targets of the considered compound (5-flourouracil, 5FU). Subsequently the basal expression of genes in the selected pathways, observed in the considered organoid biobanks, are analysed via penalised linear regression in order to identify signatures of genes predictive of 5FU response.

Following this, few identified predictive pathway signatures are successfully validated using data generated on purpose from isogenic cell lines and public clinical data.

Briefly, the approach described in this manuscript tackles a critical and timely problem. Assembling models predictive of drug response in cancer patients from public preclinical data is in fact of paramount importance for personalised medicine.

Despite the quite convincing experimental validation on the isogenic models, the analytical approach presented in this manuscript is outdated and simplistic, with several publications describing similar strategies already on the public domain, some of which using much more advanced deep learning methods (PMID: 30742607, PMID: 27993792, PMID: 31265947, PMID: 31510700, just to mention a few).

In order to show that their method represent a significant advancement with respect to previously published works the authors should have convincingly shown:

(i) That preliminary PPI guided feature selection produces indeed models with higher predictive ability with respect to just running a penalised linear regression using all the transcripts as potential regressors. This is attempted by the authors but showcasing results from a unique drug response prediction, doesn't look convincing at all to me. The authors might have performed this validation analysis using much more comprehensive public pharmacogenomics datasets, encompassing thousands of observations across hundreds of different compounds (such as for example the GDSC [PMID: 23180760] and Cancer Cell Line Encyclopaedia [PMID: 31068700])

(ii) That the selection strategy is actually effective, i.e. that regressors sets selected within pathways that are proximal to the drug targets produces indeed models with higher predictive ability with respect to just pre-selecting the same number of random features; Comparison with a 'network-free' method is not enough, what about selecting pathway in a different way? for example 'central pathway' in the ppi network regardless their proximity to the drug target

(iii) That indeed applying their method to data from the pharmacogenomic characterisation of organoids biobanks produces models with higher predictive ability with respect to using pharmacogenomic data from large-scale cell line based screens (such as the GDSC [PMID: 23180760] and Cancer Cell Line Encyclopaedia [PMID: 31068700], for which much larger collection of drug response observations (encompassing the compounds analysed in this study) are available

(iv) that this approach works better than several other network-guided regression strategies applied to the problem of predicting anti-cancer drug response, such as those cited above.

in addition, the authors do not account for possible biases arising from the reference network topology, which might contain over-characterised genes and pathways, which will be more likely to

be 'proximal' to any node due to their increased number of connections with respect to the background. A proper randomisation strategy should have been designed and implemented to estimate this possible bias, i.e. are there pathways and/or nodes that are always proximal to any target?

Finally, a possible advantage of the method presented in this manuscript over others methods might be represented by a higher level of interpretability of the selected predictive models with respect to purely data driven methods. Nevertheless the authors fail to show this.

Minor points:

- * manuscript writing style could be improved. Some sentences are not clear, grammatically incorrect and/or contain odd repetitions.
- * In the introduction, the sentence <<drugs' mechanisms-of-action were frequently proximal to drug targets>> is not clear and should be reformulated. What is the meaning of drug mode of action proximity? when it is 'proximal' to a given node?
- * Still in the introduction, 'drug target-proximal pathway' should be defined at their first occurrence
- * The the PPI reference network source together with basic information such as total number of nodes, edges and maybe density might be reported when the PPI network is mentioned for the first time.
- * The sentence <<Finally, pathways with high predictive performance or high drug dependency in the 112 predictive model were deemed predictive drug response biomarkers>> should be rewritten, in addition, it is not clear what 'high drug dependency' refer to here nor why 'high predictive performances' alone is not enough.

Reviewer #2:

Remarks to the Author:

--In this manuscript, the authors convincingly demonstrate the power of combining a network approach with pharmacogenomics (gene expression) for identifying biomarkers that predict therapeutic response in colorectal and bladder cancer patients.

--Emphasize that pathway proximity to drug targets reflect both 'upstream' and 'downstream' relationships. As the protein-protein interaction network is not a directed network, potential drug effects (i.e., beneficial vs. adverse) on pathways (or pathway prediction of drug effects) can only be ascertained experimentally.

--Please analyze the predictive differences of ridge regression vs. linear regression vs. support vector regression using as 'ground truth' patient outcomes (tumor response, survival/death) and determining the AUC of the resulting ROCs. In addition, to sort out whether there is overlap vs. complementarity among these methods, you may wish to consider utilizing a Kendall tau statistic across the three pipelines. Should there be complementarity, you might consider creating a rank aggregation algorithm and calculating a combined ROC.

Reviewer #3:

Remarks to the Author:

Kim et al. present a timely integrative systems biology based analysis of pharmacogenetic data from organoids and transcriptomics data and drug response data from patients to identify drug response pathways that have predictive potential as biomarkers of survival across patients in colorectal and bladder cancer. The manuscript is well organized, clearly written and supports the main findings of the authors. That being said, in my opinion, several issues I highlight below need to be addressed to strengthen its soundness.

1. The analysis of data from organoids to identify biomarkers that could predict drug response in patients is very interesting and builds up nicely on the recent literature. Some additional relevant articles could be included in the introduction:
 - a. Ooft et al. STM, 2019, DOI: 10.1126/scitranslmed.aay2574
 - b. Phan et al. Commun Biol, DOI: 10.1038/s42003-019-0305-x
2. In the first section of results where the authors provide an overall summary of the platform, it would help to include more info on the two machine learning models that has been tried and the number of samples for each cancer type.
3. It has been shown in the literature that random signatures have also certain predictive capacity. Accordingly, the potential overfitting issues and robustness of signatures should be checked carefully. I understand that the authors use cross validation for choosing parameters of the regression classifiers however in my opinion this is still prone to overfitting. Several experiments I could suggest in that regards are
 - a. Splitting the data from organoids into two and see whether similar pathways appear in each subset or splitting data into training (60%), validation (10%) and test set (30%)
 - b. What would be the predictive accuracy if the authors were to use activation of BH3 only proteins pathway for bladder cancer and amino acid synthesis and interconversion pathway for colorectal cancer. This would help associating certain specificity to these pathways.
4. It is interesting to see that the pathways change significant expression changes between sensitive and resistant cell lines. The resistance to threatment is certainly an important aspect that deserves further attention. I would recommend the authors provide the list of top 3-5 proximal pathways and their predictive capacity in addition to only the best candidate they have discussed in the text. In that regards, they could also exploit the two treatments capacity to target multiple (top) pathways simultaneously using a method such as PXEA (Aguirre-Plans et al., Pharmaceuticals, 2018, DOI: 10.3390/ph11030061).
5. In relation to my previous comment #3, I find it the intriguing that there were many cases in the random analysos that some pathway was predictive of the drug response. I think this deserves a bit more discussion. It would also be interesting to see whether there is a correlation / bias with respect to the number of genes a pathway has and its being predictive of drug response.
6. In all statistical tests reported in the text, the statistical test, effect sizes (e.g., confidence intervals) and the obtained P-values should be specified (instead of $P < 0.05$). The alpha cutoff used for significance should be specified clearly on the Methods. Also on the figures, it would help to have number of samples (n) whenever applicable.

Minor:

- "drugs' mechanism-of-action were frequently proximal to drug targets" unclear, mechanism-of-action related protein/pathways?

RESPONSE to REVIEWER COMMENTS

Reviewer 1.

The authors present an approach to identify transcriptional signatures predictive of drug response via a network based analysis of public available pharmacogenomic data, from the characterisation of biobanks of colorectal and bladder cancer organoids.

To this aim, the authors use a statistical strategy to identify pathway gene sets that are more proximal than expected by random chance to the nominal targets of the considered compound (5-flourouracil, 5FU). Subsequently the basal expression of genes in the selected pathways, observed in the considered organoid biobanks, are analysed via penalised linear regression in order to identify signatures of genes predictive of 5FU response.

Following this, few identified predictive pathway signatures are successfully validated using data generated on purpose from isogenic cell lines and public clinical data.

Briefly, the approach described in this manuscript tackles a critical and timely problem. Assembling models predictive of drug response in cancer patients from public preclinical data is in fact of paramount importance for personalised medicine.

Despite the quite convincing experimental validation on the isogenic models, the analytical approach presented in this manuscript is outdated and simplistic, with several publications describing similar strategies already on the public domain, some of which using much more advanced deep learning methods (PMID: 30742607, PMID: 27993792, PMID: 31265947, PMID: 31510700, just to mention a few).

In order to show that their method represent a significant advancement with respect to previously published works the authors should have convincingly shown:

(i) That preliminary PPI guided feature selection produces indeed models with higher predictive ability with respect to just running a penalised linear regression using all the transcripts as potential regressors. This is attempted by the authors but showcasing results from a unique drug response prediction, doesn't look convincing at all to me. The authors might have performed this validation analysis using much more comprehensive public pharmacogenomics datasets, encompassing thousands of observations across hundreds of different compounds (such as for example the GDSC [PMID: 23180760] and Cancer Cell Line Encyclopedic [PMID: 31068700])

To demonstrate the predictive ability of the protein-protein interaction (PPI)-guided feature selection method in the organoid datasets, we also tested the predictive ability using expression levels of 1) all the pathways and 2) the network neighbors of drug targets in a PPI network (Fig. 3b-d). Moreover, we made a comparison of all the suggestions made by the reviewer, which includes 1) selection of the same number of features using network central pathways and 2) a comparison with previously published drug response prediction methods, including deep-learning algorithms (Fig. 3e, f, Supplementary Figs. 6, 7). We found that all of the methods were less predictive compared to our method, suggesting that the PPI-guided feature selection has predictive ability in drug response. We discuss the details of the compared methods in the following comment sections (comments 2, 4). We also performed the analysis using comprehensive public pharmacogenomic data of cancer cell lines (GDSC) (comment 3).

(ii) That the selection strategy is actually effective, i.e. that regressors sets selected within pathways that are proximal to the drug targets produces indeed models with higher predictive ability with respect to just pre-selecting the same number of random features; Comparison with a 'network-free' method is not enough, what about selecting pathway in a different way? for example 'central pathway' in the ppi network regardless their proximity to the drug target

We thank the reviewer for the excellent suggestion. We agree that due to the scale-free nature of PPI networks (i.e., a network topology with very few highly connected proteins), predictive features may be identified solely from network central pathways. **We now show that the use of network central pathways to train organoid models is not predictive of**

the patient responses to 5-fluorouracil treatment for colorectal cancer and cisplatin treatment for bladder cancer. In summary, we used three different measures (degree, betweenness, and closeness centrality) to calculate centrality of each gene in the PPI network. To define the representative centrality score for each pathway, we computed the average of the centrality levels within the genes in the pathway. Next, for each centrality measurement, we selected the top central pathways, while matching the number of proximal pathways. These top central pathways were then tested against organoid pharmacogenomic data for ridge regression training. Similar to our approach using proximal pathways, we selected the top central pathways with high predictive performances (the top three pathways), and then the pathways were used to predict the drug responses of cancer patients. We found that none of the selected pathways were predictive of survival (log-rank test; $P < 0.05$ was considered significant), suggesting that centrality alone is not sufficient to identify robust drug response biomarkers (Supplementary Figs. 6, 7). Furthermore, we also showed that random selection of the same number of pathways with proximal pathways, regardless of network topology, was not predictive of patient survival and drug resistance in isogenic cancer cell lines (Fig. 4d–f). All of these results suggest that neither network centrality nor random feature selection are able to capture the signals similar to our method.

The above results have now been added to the manuscript (page 11).

Supplementary Fig. 6. Prediction of drug response in 5-fluorouracil-treated colorectal cancer patients using central pathways to train colorectal cancer organoid models. Predictive performances of the top central pathways measured by (a) degree, (b) betweenness, and (c) closeness centrality. Kaplan-Meier survival plot for the top first (yellow), second (blue), and third (red) drug response predictive pathways. The average centrality scores of pathway genes were used to quantify pathway centrality.

Supplementary Fig. 7. Prediction of drug response in cisplatin-treated bladder cancer patients using central pathways to train bladder cancer organoid models. Predictive performances of the top central pathways measured by (a) degree, (b) betweenness, and (c) closeness centrality. Kaplan-Meier survival plot for the top first (yellow), second (blue), and third (red) drug response predictive pathways. The average centrality scores of pathway genes were used to quantify pathway centrality.

(iii) That indeed applying their method to data from the pharmacogenomic characterisation of organoids biobanks produces models with higher predictive ability with respect to using pharmacogenomic data from large-scale cell line based screens (such as the GDSC [PMID: 23180760] and Cancer Cell Line Encyclopaedia [PMID: 31068700], for which much larger collection of drug response observations (encompassing the compounds analysed in this study) are available

The physiological representation of primary tumors by preclinical models is very important in the field of drug response screening/prediction. As suggested by the reviewer, we tested our method on cancer cell line datasets to measure their predictive ability. **We found that cancer cell line datasets were not strong predictors of drug responses in 5-fluorouracil-treated colorectal and cisplatin-treated bladder cancer patients (Supplementary Fig. 15).** We used GDSC cancer cell line datasets to predict the drug responses of cancer patients, which included drug screening data for both 5-fluorouracil and cisplatin. Because several previous studies suggested that the use of all cancer cell lines, regardless of their tissue origin, was often more predictive than the use of only tissue type-specific cancer cell lines¹, we separately tested both tissue type-specific and all available cancer cell lines to train the machine-learning models.

Of note, for drug-sensitive and -resistant isogenic cancer cell lines, our network-based approach identified the pathways that were clearly distinctive in gene expression (Fig. 4a–c). This strongly suggests that it is crucial to select cancer cell lines with high resemblance to primary tumors to accurately predict drug responses^{2,3}. Indeed, Acar *et al.* recently showed that re-plated cancer cell line cultures were susceptible to stochastic, unpredictable drug resistance evolution⁴, which could hinder the performance of all downstream processes.

We have now provided a discussion to address this point in the manuscript (page 19).

Supplementary Fig. 15. Drug response predictions of 5-fluorouracil-treated colorectal and cisplatin-treated cancer patients using the network-based approach in cancer cell lines from the Genomics of Drug Sensitivity in Cancer (GDSC) database. The drug response prediction of 5-fluorouracil in colorectal cancer patients using (a) all cancer cell lines (n = 893) and (b) colorectal cancer cell lines (n = 46). The drug response prediction of cisplatin in bladder cancer patients using (c) all cancer cell lines (n = 831) and (d) bladder cancer cell lines (n = 14). *P*-values were calculated using the log-rank test.

(iv) that this approach works better than several other network-guided regression strategies applied to the problem of predicting anti-cancer drug response, such as those cited above.

We thank the reviewer for pointing out these interesting references. As suggested by the reviewer, we first applied the method by Bolis *et al.*⁵ (PMID: 27993792) to our dataset to predict the drug responses of cancer patients. **We found that the method was less powerful for drug response prediction compared to our method, using both the colorectal and bladder cancer datasets (Fig. 3e).** In brief, Bolis *et al.* reduced the number of features by correlating drug response and gene expression from cancer cell line data, using 10 iterations of leave-half-out cross-validation. We implemented this step into the organoid datasets used in our manuscript, and the selected features were used to train the machine-learning models. The trained machine-learning models were not predictive of cancer patient survival. The log-rank test *P*-values were 0.15 and 0.96 for 5-fluorouracil-treated colorectal and cisplatin-treated bladder cancer patients, respectively. The reduced performances may be due to the smaller number of samples used to select input features and to train the machine-learning models. We would also like to note that we were unable to implement the second feature selection step conducted by Bolis *et al.*, which removed genes that were missing in a human co-expression network. No features were retained after the second feature selection step, which may be due to the small number of samples in the organoid datasets. Better assessment of the method will be possible after more pharmacogenomic data of organoid datasets become available.

We also applied the deep neural network model developed by Sharifi-Noghabi *et al.*⁶ (PMID: 31510770) to the organoid data. **We observed that deep neural network models exhibited less predictive ability for drug responses (Figure 3f).** In summary, Sharifi-Noghabi *et al.* used a combined loss function using a triplet loss and binary cross-entropy loss to optimize the prediction model. By training the transcriptomic data of the organoid models via the deep neural network model, the predicted overall survival for both drug-cancer types were statistically insignificant, with log-rank *P*-values of 0.91 and 0.82 for colorectal cancer after 5-fluorouracil treatment and bladder cancer after cisplatin treatment, respectively. In

addition to the reduced predictive ability, we would also like to point out that the deep neural network proposed by Sharifi-Noghabi *et al.* was designed to use multi-omic data as inputs for the machine-learning model. In our study, we only focused on transcriptomic data because we wanted to identify gene expression biomarkers. We agree with the authors that incorporating multi-omic data can improve performances in drug response prediction. Indeed, in our study we showed the consistency between mutational signatures and expression-based biomarkers (Figure 5), further supporting the idea that using multi-omic datasets may improve predictive performances. Nonetheless, we would still like to know whether the incorporation of multi-omic data into our network-based method improves the predictive ability, but this will only be possible when large-scale multi-omic screening data of organoid models become available.

We have now updated the results in the manuscript (page 11, lines 6–23).

Fig. 3. Drug response predictions using other machine-learning methods. (a) Overall scheme of other drug response prediction methods using 1) no feature selection and 2) a feature selection process. (b)–(d) Survival predictions were trained using (b) whole-genome profiles, (c) whole-pathway profiles, (d) first neighbors of drug targets, (e) feature selection (Bolis *et al.*), and (f) a deep-learning method (Sharifi-Noghabi *et al.*). Kaplan-Meier survival plots and log-rank test P -values are displayed.

in addition, the authors do not account for possible biases arising from the reference network topology, which might contain over-characterised genes and pathways, which will be more likely to be ‘proximal’ to any node due to their increased number of connections

with respect to the background. A proper randomisation strategy should have been designed and implemented to estimate this possible bias, i.e. are there pathways and/or nodes that are always proximal to any target?

We thank the reviewer for paying close attention to the analytical details of our research. Biased outcomes toward central genes or pathways in a PPI network is indeed one of the major problems in the field of network science. Therefore, we took extra caution in selecting a method to calculate the proximity between drug targets and pathways. We used the network proximity measurements provided in a report by *Guney et al.*⁷ (2016) (PMID: 26831545), in which the authors explicitly showed that their proximity measurements exhibited no correlation with the network degree (Fig. 2b, c). Furthermore, as suggested by the reviewer, we examined if any of the pathways were always proximal to the drugs in the PPI network that we used in our study. **We found that no pathway was proximal to all of the drug targets, suggesting that randomized proximity measurement successfully reduced the biased dependency toward central genes/pathways.** To test for potential biased dependency, we tested 44 drugs that were screened in the colorectal (PMID: 25957691) and bladder cancer organoid datasets (PMID: 29625057), and we measured the proximity between the drug targets and pathways. As shown in Supplementary Fig. 16, we found that only 3 out of 674 pathways (0.45%) were proximal to more than half of the drugs tested (proximal to 22 drugs or more), whereas a pathway was proximal to an average of 4.9 drugs (median: 4 drugs per pathway). This result suggests that randomized process was useful to reduce the potential bias toward any network central pathways.

Supplementary Fig. 16. Number of proximal pathways with the drug targets of 44 drugs.

The number of proximal pathways ($z\text{-score} \leq -1.2816$) for each drug (44 drugs total) is shown. The total numbers of drugs and pathways are shown inside the parentheses. Pathways that are not proximal to any of the 44 drugs are not displayed ($n = 111$ pathways).

Finally, a possible advantage of the method presented in this manuscript over others methods might be represented by a higher level of interpretability of the selected predictive models with respect to purely data driven methods. Nevertheless the authors fail to show this.

We thank the reviewer for this excellent suggestion. We have now included the following paragraph in the Discussion section of our manuscript:

“Interpretability of ML models is becoming increasingly important, especially when it comes to the high-stakes decision making of healthcare⁴⁰. To generate interpretable ML models, incorporation of biological structures, such as PPI networks, into the ML models assists in making comprehensive predictions^{10,11}. Recently, Ma *et al.* showed that the infusion of hierarchical structures of biological functions into neural networks accurately predicted cellular growth from genotypes⁴¹. Unlike standard deep-learning methods, interpretable

prediction results provide information on the underlying biological mechanisms of genotype-phenotype relationships^{10,41} which can be experimentally validated and leveraged for the development of novel, personalized, targeted therapies or for an increased understanding of a disease.”

Minor points:

** manuscript writing style could be improved. Some sentences are not clear, grammatically incorrect and/or contain odd repetitions.*

We have had our manuscript professionally edited by BioScience Writers, LLC to improve the language and writing style.

** In the introduction, the sentence <<drugs’ mechanisms-of-action were frequently proximal to drug targets>> is not clear and should be reformulated. What is the meaning of drug mode of action proximity? when it is ‘proximal’ to a given node?*

As suggested by the reviewer, we have updated the sentence in the manuscript (page 3, line23) :

“Fernández-Torras *et al.* showed that gene modules in a network could be used to predict drug response.”

** Still in the introduction, ‘drug target-proximal pathway’ should be defined at their first occurrence*

As pointed out by the reviewer, we have included the following sentence in the text (page 4, line22):

“Specifically, prior to ML, we conducted a feature selection procedure in which features (pathways) were selected by their proximity to drug targets in a PPI network.”

** The PPI reference network source together with basic information such as total number of nodes, edges and maybe density might be reported when the PPI network is mentioned for the first time.*

We have included the number of nodes and edges in the first paragraph of the Results section:

“Here, we used the STRING PPI network, which is comprised of 13,824 proteins and 323,774 interactions.”

** The sentence <<Finally, pathways with high predictive performance or high drug dependency in the 112 predictive model were deemed predictive drug response biomarkers>> should be rewritten, in addition, it is not clear what ‘high drug dependency’ refer to here nor why ‘high predictive performances’ alone is not enough.*

As suggested by the reviewer, we have changed the expression “drug dependency” to “predictive performance” throughout the entire manuscript.

Reviewer 2.

--In this manuscript, the authors convincingly demonstrate the power of combining a network approach with pharmacogenomics (gene expression) for identifying biomarkers that predict therapeutic response in colorectal and bladder cancer patients.

--Emphasize that pathway proximity to drug targets reflect both ‘upstream’ and ‘downstream’ relationships. As the protein-protein interaction network is not a directed network, potential drug effects (i.e., beneficial vs. adverse) on pathways (or pathway prediction of drug effects) can only be ascertained experimentally.

We thank the reviewer for this suggestion because we did indeed use an undirected PPI

network, which may cause a misinterpretation of the results if not clearly stated. As suggested by the reviewer, **we have now included the following sentence in the text (page 6 lines 6–7).**

“Of note, the STRING network is an undirected network and therefore does not have “upstream” or “downstream” directionality.”

--Please analyze the predictive differences of ridge regression vs. linear regression vs. support vector regression using as ‘ground truth’ patient outcomes (tumor response, survival/death) and determining the AUC of the resulting ROCs. In addition, to sort out whether there is overlap vs. complementarity among these methods, you may wish to consider utilizing a Kendall tau statistic across the three pipelines. Should there be complementarity, you might consider creating a rank aggregation algorithm and calculating a combined ROC.

We thank the reviewer for this comment. To address the question raised by the reviewer, we compared the regression coefficients from various machine-learning models to compare the potential differences in the generation of drug response biomarkers. We found that the predictive abilities of the machine-learning models were highly similar. Specifically, we trained organoid models using the transcriptomes of drug target proximal pathways and IC₅₀ values. To quantify similarity (or disparity) among the three different machine-learning models, we tested the pairwise correlation of regression coefficients. We observed high correlations ($R^2 > 0.99$) from all of the pairwise correlations tested (Supplementary Fig. 3), suggesting that same biomarkers were selected regardless of the machine-learning algorithms. Additionally, the regression coefficients from the three different algorithms were nearly identical, suggesting that there were almost no complementary effects when combining the different methods.

Supplementary Fig. 3. Comparison of regression coefficients between various machine-learning algorithms. Various machine-learning models were trained using the transcriptomes of proximal pathways and drug responses (IC_{50} values) from organoid models. Correlation of regression coefficients between machine-learning algorithms in (a)–(c) for 5-fluorouracil-treated colorectal cancer organoids and in (d)–(e) for cisplatin-treated bladder cancer organoids. The coefficient of determination is given as R^2 . SVR, support vector regression.

Reviewer 3.

Kim et al. present a timely integrative systems biology based analysis of pharmacogenetic data from organoids and transcriptomics data and drug response data from patients to identify drug response pathways that have predictive potential as biomarkers of survival across patients in colorectal and bladder cancer. The manuscript is well organized, clearly written and supports the main findings of the authors. That being said, in my opinion, several issues I highlight below need to be addressed to strengthen its soundness.

1. The analysis of data from organoids to identify biomarkers that could predict drug response in patients is very interesting and builds up nicely on the recent literature. Some additional relevant articles could be included in the introduction:

a. Ooft et al. STM, 2019, DOI: 10.1126/scitranslmed.aay2574

b. Phan et al. Commun Biol, DOI: 10.1038/s42003-019-0305-x

We thank the reviewer for introducing these interesting references. **The suggested references have now been included in the Introduction section (page 4).**

“Similarly, Ooft *et al.* discovered that organoid models were predictive of colorectal cancer patient responses to irinotecan-based chemotherapy¹⁹. Thus, because of the high similarity between organoid models and human tumors, methods to culture and screen organoid models in an automated, high-throughput manner are actively being developed²⁰.”

2. In the first section of results where the authors provide an overall summary of the platform, it would help to include more info on the two machine learning models that has been tried and the number of samples for each cancer type.

We have followed the advice given by the reviewer and **added the suggestions in the first section of the Results (page 6).**

3. It has been shown in the literature that random signatures have also certain predictive capacity. Accordingly, the potential overfitting issues and robustness of signatures should be checked carefully. I understand that the authors use cross validation for choosing parameters of the regression classifiers however in my opinion this is still prone to overfitting. Several experiments I could suggest in that regards are

a. Splitting the data from organoids into two and see whether similar pathways appear in each subset or splitting data into training (60%), validation (10%) and test set (30%)

We thank the reviewer for this suggestion, and we agree that overfitting issues could have been more thoroughly investigated. Following the reviewer’s suggestion, we split the data into training (60%), validation (10%), and test (30%) sets and measured the model’s

predictive ability. **We observed high correlation between the observed and predicted drug responses in our test sets.** Specifically, we split organoid datasets into test sets (COAD, n = 6; BLCA, n = 3) and used the rest as training (COAD, n = 11; BLCA, n = 5) or validation (COAD, n = 2; BLCA, n = 1) sets. After splitting the samples, we used expression profiles of drug target proximal pathways to train the machine-learning model. In detail, we used ridge regression to train the organoid pharmacogenomic data, and training and validation sets were used to optimize the hyperparameter of the regression model (α was between 0.1–1 using a 0.1 interval). We then used the optimized hyperparameter to predict drug responses of the test sets. Performance of the predictions were measured by the correlation between the observed and predicted IC_{50} values of the test set. To test whether the specific validation set had an effect on the predictions, we sampled all possible combinations of the validation set within the training set to predict drug responses. We found high correlation between the observed and predicted drug responses, and the correlation coefficients (R^2) were 0.98 for the colorectal and 0.89 for bladder cancer organoid models (Supplementary Fig. 17). These results suggest that a machine-learning model trained using the transcriptomes of proximal pathways is able to predict robust drug responses.

Supplementary Fig. 17. Prediction of drug response in a split organoid dataset. Organoid samples were split into training (60%), validation (10%), and test (30%) sets. Prediction performance was measured by comparing the observed and predicted drug responses in (a) 5-

fluorouracil-treated colorectal and **(b)** cisplatin-treated bladder cancer organoids. Samples used for training and test sets are indicated as yellow and green dots, respectively. Scatter plot shows the 95% confidence interval.

b. What would be the predictive accuracy if the authors were to use activation of BH3 only proteins pathway for bladder cancer and amino acid synthesis and interconversion pathway for colorectal cancer. This would help associating certain specificity to these pathways.

We thank the reviewer for this excellent suggestion. As suggested by the reviewer, we inter-crossed the biomarkers found from the colorectal and bladder cancer organoids and measured their predictive ability. **We found that the predictive performances were not strong when inter-crossing the biomarkers.** In detail, the “amino acid synthesis and interconversion” pathway for 5-fluorouracil-treated colorectal cancer had a log-rank P -value of 0.16, and the “activation of BH3-only proteins” pathway for cisplatin-treated bladder cancer had a P -value of 0.86 (Supplementary Fig. 4), suggesting that the identified biomarkers are specific to the drug and cancer type. **We have now included this result in the text (page 9).**

Supplementary Fig. 4. Identification of biomarkers across drug and cancer types. (a) The cisplatin biomarker in bladder cancer (“amino acid synthesis and interconversion” pathway) was used to classify 5-fluorouracil-treated colorectal cancer patients. **(b)** The 5-fluorouracil biomarker in colorectal cancer (“activation of BH3-only proteins” pathway) was used to classify cisplatin-treated bladder cancer patients.

4. It is interesting to see that the pathways change significant expression changes between sensitive and resistant cell lines. The resistance to treatment is certainly an important aspect that deserves further attention. I would recommend the authors provide the list of top 3-5 proximal pathways and their predictive capacity in addition to only the best candidate they have discussed in the text. In that regards, they could also exploit the two treatments capacity to target multiple (top) pathways simultaneously using a method such as PxEA (Aguirre-Plans et al., Pharmaceuticals, 2018, DOI: 10.3390/ph11030061).

We thank the reviewer for this great suggestion. As suggested by the reviewer, we have now provided the top three proximal pathways and their predictive capacities in drug-sensitive and -resistant cancer cell lines (Supplementary Table 3). We have also tested whether the predictive performances for the drug responses of cancer patients could increase when simultaneously using multiple pathways. **We found that the predictive performance was maintained for cisplatin-treated bladder cancer patients when using the top two pathways, whereas no predictive advantage was observed when adding more pathways to predict the drug responses of 5-fluorouracil-treated colorectal cancer patients.** To enable the prediction model that utilized multiple pathways, we calculated the weighted sum of the pathway expression levels for patients, and the expression of a pathway was weighted by the predictive ability of the pathway in the organoid models (see Methods, equation (2)). We found that when measuring the predictive performances using the top two pathways with the highest predictive ability in the organoid models, the log-rank test gave *P*-values of 0.88 and 0.022 for 5-fluorouracil-treated colorectal and cisplatin-treated bladder cancer patients, respectively (Supplementary Fig. 5). We provided the measured predictive performances using up to the top 10 pathways in the Source Data file. Our results suggest that the treatment capacities to target multiple pathways may differ across the drugs and cancer types; however, further analyses in comprehensive datasets are warranted. One interesting future research objective would be to couple individual patient omic information with tools, such as PxEA¹⁰, which identifies drugs that target multiple disease-related pathways for personalized medicine.

Cancer	Drug	Pathway	Proximity (z)	Mean. Sensitive	Mean. Resistant	P-value
COAD	5FU	MITOTIC_G1_G1_S_PHASES	-4.26	0.34	0.31	0.15
		G1_S_TRANSITION	-3.9	0.38	0.34	0.15
		PYRIMIDINE_METABOLISM	-3.42	0.17	0.16	0.11
BLCA	Cisplatin	METAL_ION_SLC_TRANSPORTERS	-4.03	0.078	0.06	0.014
		RESPONSE_TO_ELEVATED_PLATELET_CYTOSOLIC_CA2	-3.75	0.14	0.12	0.00054
		RECYCLING_OF_BILE_ACIDS_AND_SALTS	-3.73	-0.28	-0.22	0.01

Supplementary Table 3. Expression differences of the top three proximal pathways between drug-sensitive and -resistant isogenic cancer cell lines. The top three most proximal pathways to 5-fluorouracil and cisplatin treatment were used to measure expression level differences. Network proximity (z-scores), mean pathway expression levels in drug-sensitive and -resistant cancer cell lines, and *P*-values for expression level differences are displayed. The Student's t-test was used to calculate the *P*-values.

Supplementary Fig. 5. Prediction of drug response in cancer patients using the two pathways with the highest predictive performances in organoid models. Drug response prediction for (a) 5-fluorouracil-treated colorectal and (b) cisplatin-treated bladder cancer patients.

5. In relation to my previous comment #3, I find it the intriguing that there were many cases in the random analysis that some pathway was predictive of the drug response. I think this deserves a bit more discussion. It would also be interesting to see whether there is a correlation / bias with respect to the number of genes a pathway has and its being predictive of drug response.

Following the reviewer's suggestion, we have now provided all of the pathways that were predictive of drug responses in the random analysis (Supplementary Table 4). We found that other than biomarker pathways discovered from our analysis, one pathway in 5FU-treated colorectal cancer and six pathways in cisplatin-treated bladder cancer were predictive of the

drug responses in the analysis. Also, we observed that there was no correlation between the number of genes in a pathway and their predictive performances for the drug responses of cancer patients (Supplementary Fig. 10). We found that nearly no correlation existed in either 5FU-treated colorectal ($R^2 = 0.0026$) or cisplatin-treated bladder ($R^2 = 0.00034$) cancer, suggesting that the number of genes in a pathway was not a confounding factor in our prediction analysis.

Cancer	Drug	Pathway	Proximity (z)
COAD	5FU	PI3K_EVENTS_IN_ERBB2_SIGNALING	1.4
BLCA	Cisplatin	REGULATION_OF_KIT_SIGNALING	-1.51
		NOTCH1_INTRACELLULAR_DOMAIN_REGULATES_TRANSCRIPTION	-0.47
		SIGNALING_BY_NOTCH1	-0.16
		ADAPTIVE_IMMUNE_SYSTEM	1.63
		IL_2_SIGNALING	2.09
		CA_DEPENDENT_EVENTS	2.94

Supplementary Table. 4. Network proximity for predictive pathways of drug response from bootstrapping analysis. Pathways that ranked equal to or higher than our predictions in the bootstrapping analysis are displayed.

Supplementary Fig. 10. Correlation between the number of genes in a pathway and its predictive performance in cancer patient drug responses. Correlation between the number of genes in a pathway and drug response prediction in (a) 5-fluorouracil-treated colorectal and (b) cisplatin-treated bladder cancer patients. Patients were divided into two groups according to the median IC_{50} value and using pathway expression levels.

6. In all statistical tests reported in the text, the statistical test, effect sizes (e.g., confidence intervals) and the obtained P-values should be specified (instead of $P < 0.05$). The alpha cutoff used for significance should be specified clearly on the Methods. Also on the figures, it would help to have number of samples (n) whenever applicable.

As the reviewer has suggested, we have made the following changes in the current manuscript: 1) confidence intervals for Kaplan-Meier survival plots have been provided, 2) *P*-values have been specified throughout the text, 3) the α value used for network proximity cut-off has been included, and 4) the sample sizes have been added to the figures. We have also included the sample distributions for Fig. 4.

Minor:

- “drugs’ mechanism-of-action were frequently proximal to drug targets” unclear, mechanism-of-action related protein/pathways?

We thank the reviewer for this suggestion, which was also mentioned by Reviewer 1. We have now clarified the sentence (page 3) :

“Fernández-Torras *et al.* showed that gene modules in a network could be used to predict drug response.”

References

1. Geeleher, P., Cox, N. J. & Huang, R. Clinical drug response can be predicted using baseline gene expression levels and in vitro drug sensitivity in cell lines. *Genome Biol.* **15**, R47 (2014).
2. Liu, K. *et al.* Evaluating cell lines as models for metastatic breast cancer through integrative analysis of genomic data. *Nat. Commun.* (2019). doi:10.1038/s41467-019-10148-6
3. Yu, K. *et al.* Comprehensive transcriptomic analysis of cell lines as models of primary tumors across 22 tumor types. *Nat. Commun.* (2019). doi:10.1038/s41467-019-11415-2
4. Acar, A. *et al.* Exploiting evolutionary steering to induce collateral drug sensitivity in cancer. *Nat. Commun.* **11**, 1923 (2020).
5. Bolis, M. *et al.* Network-guided modeling allows tumor-type independent prediction of sensitivity to all-trans-retinoic acid. *Ann. Oncol.* **28**, 611–621 (2017).
6. Sharifi-Noghabi, H., Zolotareva, O., Collins, C. C. & Ester, M. MOLI: multi-omics late integration with deep neural networks for drug response prediction. *Bioinformatics* **35**, i501–i509 (2019).
7. Guney, E., Menche, J., Vidal, M. & Barábasi, A.-L. Network-based in silico drug efficacy screening. *Nat. Commun.* **7**, 10331 (2016).
8. Ooft, S. N. *et al.* Patient-derived organoids can predict response to chemotherapy in metastatic colorectal cancer patients. *Sci. Transl. Med.* **11**, eaay2574 (2019).
9. Phan, N. *et al.* A simple high-throughput approach identifies actionable drug sensitivities in patient-derived tumor organoids. *Commun. Biol.* **2**, 78 (2019).
10. Aguirre-Plans, J. *et al.* Proximal Pathway Enrichment Analysis for Targeting Comorbid Diseases via Network Endopharmacology. *Pharmaceuticals* **11**, 61 (2018).

Reviewers' Comments:

Reviewer #1:

Remarks to the Author:

The authors performed an excellent work to address mine and others' reviewers concerns. To this aim they included several additional analysis and results now convincingly showing that their analytical platform and data outperform existing methods and their results are not biased by the reference network topology.

The two final minor points might be discretionarily addressed by the authors, before proceeding with the manuscript acceptance:

* in figure 2c and 2g, some of the node labels are not readable. I would use a bold-face font or add a white rectangle as a label background

* while mentioning the importance of selecting cancer cell lines with high resemblance to primary tumors in page 11, the following work might be cited: PMID: 32437684

Reviewer #2:

Remarks to the Author:

N/A

Reviewer #3:

Remarks to the Author:

I thank authors for meticulously addressing the points raised by me and the other reviewers. I have no further reservation and recommend it for publication.

REVIEWERS' COMMENTS

Reviewer #1 (Remarks to the Author):

The authors performed an excellent work to address mine and others' reviewers concerns.

To this aim they included several additional analysis and results now convincingly showing that their analytical platform and data outperform existing methods and their results are not biased by the reference network topology.

We thank the reviewer for constructive and insightful comments.

The two final minor points might be discretionarily addressed by the authors, before proceeding with the manuscript acceptance:

** in figure 2c and 2g, some of the node labels are not readable. I would use a bold-face font or add a white rectangle as a label background*

As the reviewer's suggestion, we now have added a white rectangle as a label background for the node labels in figure 2c and 2g.

Fig. 2. Identification of biomarkers in organoid models and validation in human tumors.

(a) Network distances and z-scores of Reactome pathways for 5-fluorouracil. Proximal pathways (see Methods) are shown in blue. (b) Predictive performances of pathways from colorectal cancer organoid models. Ridge regression was performed to calculate predictive

performances. Selected drug response biomarker for clinical drug response prediction is shown in orange. (c) Network representation of 5-fluorouracil target and the biomarker pathway. (d) (Left) Drug response predictions for 5-fluorouracil-treated colorectal cancer. Predicted responders and non-responders are depicted in purple and yellow, respectively. (Right) Predictions for patients with no known treatment history are shown for colorectal cancer. Number of patients is shown inside the parenthesis. 95% confidence interval was used. Statistical significance was measured using Kaplan-Meier survival curves and two-sided log-rank tests. P -values < 0.05 were considered significant. (e)-(f) Biomarker identification from cisplatin-treated bladder cancer organoids. (g) Network representation of cisplatin targets and the biomarker pathway. (h) Drug response prediction for cisplatin-treated bladder cancer patients. Kaplan-Meier survival plots and two-sided log-rank P -values are displayed.

** while mentioning the importance of selecting cancer cell lines with high resemblance to primary tumors in page 11, the following work might be cited: PMID: 32437684*

We thank the reviewer for introducing the work. As suggested, we have added the reference (PMID: 32437684) in the corresponding discussion section.

Reviewer #2 (Remarks to the Author):

N/A

We thank the reviewer for the constructive comments during the revision process.

Reviewer #3 (Remarks to the Author):

I thank authors for meticulously addressing the points raised by me and the other reviewers. I have no further reservation and recommend it for publication.

We thank the reviewer for constructive comments and positive feedbacks.

Emre Guney